# When RAG Hurts: Diagnosing and Mitigating Attention Distraction in Retrieval-Augmented LVLMs

**Beidi Zhao** [1 2]  **Wenlong Deng** [1 2]  **Xinting Liao** [1 2]  **Yushu Li** [1 2]  **Nazim Shaikh** [3]  **Yao Nie** [* 3]  **Xiaoxiao Li** [* 1 2]

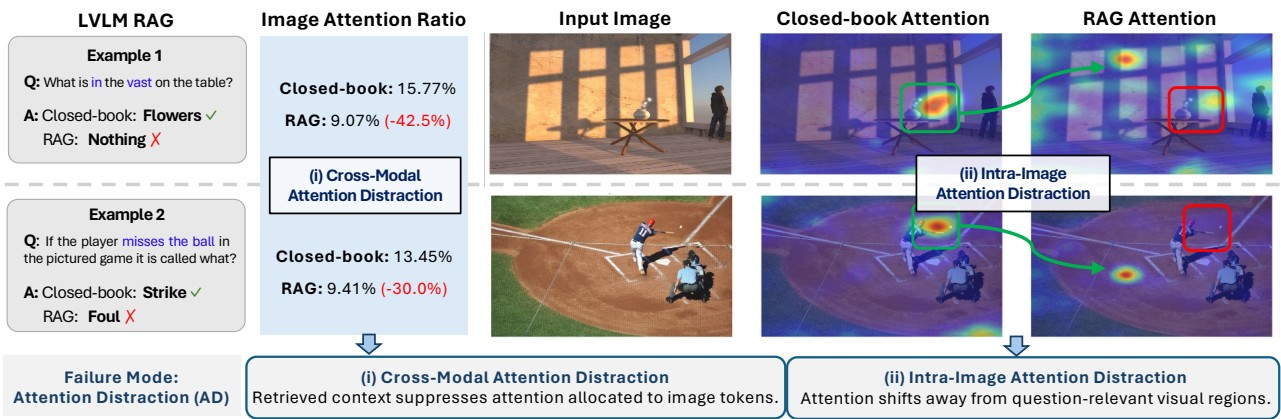

*Figure 1.* **Attention Distraction (AD) phenomenon introduced by RAG in LVLMs.** We show the average image attention ratios over all heads and visualize the attention heatmaps (details in Sec. 4) at the decoding stage. We observe a previously overlooked failure mode, i.e., AD, in retrieval-augmented LVLMs, including (i) cross-modal distraction, and (ii) intra-image attention distraction.

## Abstract

While Retrieval-Augmented Generation (RAG) is one of the dominant paradigms for enhancing Large Vision-Language Models (LVLMs) on knowledge-based VQA tasks, recent work attributes RAG failures to insufficient attention towards the retrieved context, proposing to reduce the attention allocated to image tokens. In this work, we identify a distinct failure mode that previous study overlooked: *Attention Distraction (AD)*. When the retrieved context is sufficient (highly relevant or including the correct answer), the retrieved text suppresses the visual attention globally, and the attention on image tokens shifts away from question-relevant regions. This leads to failures on questions the model could originally answer correctly without the retrieved

text. To mitigate this issue, we propose **MAD-RAG**, a training-free intervention that decouples visual grounding from context integration through a dual-question formulation, combined with attention mixing to preserve image-conditioned evidence. Extensive experiments on OK-VQA, E-VQA, and InfoSeek demonstrate that **MAD-RAG** consistently outperforms existing baselines across different model families, yielding absolute gains of up to 4.76%, 9.20%, and 6.18% over the vanilla RAG baseline. Notably, **MAD-RAG** rectifies up to 74.68% of failure cases with negligible computational overhead. Code is available at https://github.com/ubc-tea/MAD-RAG.

## 1. Introduction

Large Vision–Language Models (LVLMs) integrate high-capacity visual encoders with autoregressive language models, enabling unified reasoning over images and text (Liu et al., 2023; Bai et al., 2025). However, when visual understanding must be grounded in external factual knowledge, parametric representations alone are insufficient. In this regime, Retrieval-Augmented Generation (RAG) has emerged as the dominant framework for knowledge-based

[1]Department of Electrical and Computer Engineering, University of British Columbia, Vancouver, BC, Canada. [2]Vector Institute, Toronto, ON, Canada. [3]Roche Diagnostics, Santa Clara, CA, United States. Correspondence to: Xiaoxiao Li <xiaoxiao.li@ece.ubc.ca>, Yao Nie <yao.nie@roche.com>.

*Proceedings of the $43^{rd}$ International Conference on Machine Learning*, Seoul, South Korea. PMLR 306, 2026. Copyright 2026 by the author(s).

visual question answering (KB-VQA) (Marino et al., 2019; Mensink et al., 2023; Chen et al., 2023), supplementing LVLMs with external evidence at inference time (Lin & Byrne, 2022; Lin et al., 2024; Caffagni et al., 2025), where correct responses depend on factual information beyond the model parameters. Generally, a RAG framework comprises two primary components: a retrieval module that identifies relevant external documents and a generation module that synthesizes the retrieved information to produce responses. Prior research has dedicated substantial effort to improving these components, ranging from enhancing retrieval precision (Lin et al., 2024; Caffagni et al., 2024; 2025) to equipping generation models with capabilities to selectively leverage the retrieved knowledge (Yuan et al., 2025; Ling et al., 2025).

Despite its empirical success, RAG is not universally beneficial. Existing work offers partial perspectives but cannot account for this phenomenon in LVLMs, since it requires aligning the visual-textual modality and balancing the attention allocation of multi-modal tokens. Research on RAG calibration for LLMs (Shi et al., 2024; Yuan et al., 2024; Wang et al., 2025b; Qiu et al., 2025) focuses on balancing parametric and retrieved knowledge, but operates within the textual domain, inherently lacking the capability to address the cross-modal dynamics. More recently, ALFAR (An et al., 2026) addresses modality balance in retrieval-augmented LVLMs, arguing that models over-attend to image tokens relative to answer-relevant context tokens, and proposes re-allocating attention toward the retrieved text. This diagnosis indicates that the bottleneck lies in the under-utilization of context due to excessive visual attention. However, such a strategy overlooks the intrinsic interdependence of image and text in KB-VQA tasks, merely increasing attention to context tokens leads to suboptimal performance gains.

In this work, we identify a dinstinct failure mode. As shown in Fig. 2, RAG consistently degrades a non-trivial subset of instances that LVLMs answer correctly in the closed-book setting, even when the retrieved context is accurate and relevant. *What explains this paradox? Why does accurate retrieval cause failures on questions answerable from parametric knowledge alone?*

We term this previously overlooked failure mode in multimodal RAG as **Attention Distraction** (AD). AD refers to a systematic misallocation of attention induced by retrieved textual context during multimodal generation (as shown in Fig. 1). Specifically, retrieval augmentation alters attention dynamics in two aspects: (i) cross-modal attention distraction, where the visual attention is systematically suppressed by the retrieved context compared to using parametric knowledge, and (ii) intra-image attention distraction, where visual attention shifts from question-relevant regions to visually salient but semantically irrelevant content. Cru-

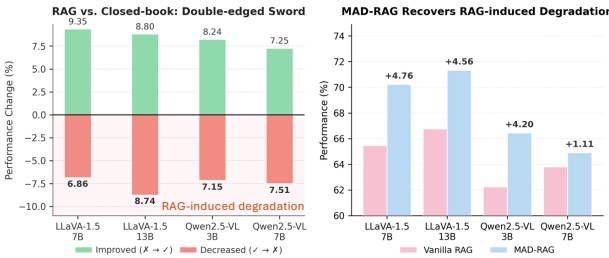

*Figure 2.* **RAG-induced degradation and recovery by MAD-RAG on OK-VQA. Left:** Sample-level performance dynamics of Vanilla RAG compared to the closed-book setting on OK-VQA. While RAG introduces gains (green), it causes some correct predictions to flip to incorrect (red), highlighting the issue of AD. **Right:** Accuracy comparison between Vanilla RAG and **MAD-RAG**. **MAD-RAG** consistently achieves higher performance by addressing the degradation issues shown on the left.

cially, AD arises even when the retrieved information is highly-related and informative, demonstrating that the degradation stems not from poor retrieval quality but from using inappropriate attention in the generation. This work focuses on the generation phase of the multimodal RAG, investigating how LVLMs internalize retrieved evidence during the decoding process.

To address AD, we propose **MAD-RAG** (**M**itigating **A**ttention **D**istraction in **R**etrieval-**A**ugmented **G**eneration for LVLMs), a training-free, inference-time intervention designed to explicitly mitigate AD in retrieval-augmented LVLMs. Motivated by an analysis of autoregressive attention dynamics under long-context multimodal inference, **MAD-RAG** addresses AD from two complementary aspects: (i) decoupling the perception to visual and external knowledge, preventing retrieved context from dominating cross-modal attention, and (ii) explicitly re-aligning attention so that retrieved context supports reasoning without suppressing image-conditioned evidence or question-relevant visual regions. Operating directly on attention dynamics, **MAD-RAG** requires no retraining or retriever modification and incurs negligible computational overhead, while substantially improving visual grounding stability under retrieval augmentation.

Our main contributions are summarized as follows:

- We characterize Attention Distraction (AD) as a distinct failure mode in retrieval-augmented LVLMs, manifesting two aspects: (i) cross-modal attention distraction, where visual attention is globally suppressed by text, and (ii) intra-image attention distraction, where focus on image shifts to irrelevant or background regions.

- We provide an empirical analysis, demonstrating that AD arises from the interaction between retrieval aug-

mentation and auto-regressive attention dynamics in long-context multimodal inference, independent of retrieval quality.

- We introduce **MAD-RAG**, a simple yet effective and efficient training-free method that robustly aligns visual attention with the question. Extensive experiments on KB-VQA datasets demonstrate consistent state-of-the-art performance, substantial recovery of attention distraction cases, and negligible inference overhead.

## 2. Related Work

### 2.1. Multi-modal RAG for Knowledge-based VQA

Recent KB-VQA advancements (Marino et al., 2019; Mensink et al., 2023; Chen et al., 2023) have shifted toward RAG to bridge knowledge gaps of LVLMs, evolving from basic retrieval (Lin & Byrne, 2022) to specialized retrievers (e.g., PreFLMR (Lin et al., 2024), Wiki-LLaVA (Caffagni et al., 2024)) and re-ranking (Yan & Xie, 2024). However, integrating external context inevitably triggers conflicts with the model's parametric knowledge (Liu et al., 2024b; Wang et al., 2025a). To address this, inference-time strategies based on logits, such as CAD (Shi et al., 2024), COIECD (Yuan et al., 2024), AdaCAD (Wang et al., 2025b), and entropy-based methods (Qiu et al., 2025), quantify uncertainty or calibrate decoding weights to balance the use of external knowledge. In multimodal extensions, ALFAR (An et al., 2026) attributes RAG failures to attention bias toward image tokens in early layers compared to answer-related context tokens and increases attention to context tokens. Complementarily, training-based methods like AlignRAG (Wei et al., 2025) and VRAG-RL (Wang et al., 2025c) utilize reinforcement learning to refine evidence sensitivity, while ReflectiVA (Cocchi et al., 2025) and MR²AG (Zhang et al., 2024) employ reflection mechanisms to calibrate internal states. However, directly applying LLM-based methods implicitly assumes that retrieval quality alone suffices for reasoning, thereby underestimating the critical role of visual evidence. Furthermore, while ALFAR considers modality balance, it merely operates on the premise of reducing visual dominance, which is not consistently applicable for the cross-modal dynamics. Crucially, none of these approaches address the specific failure mode of AD, leaving both cross-modal and intra-image distraction unresolved.

### 2.2. Visual-Grounded Reasoning in LVLMs

A line of closely related work studies visual grounding[1] in LVLMs without retrieval-augmentation. These methods

---

[1]In this context, "grounding" refers to the model's capability to anchor its generated responses in visual evidence, distinguishing it from object localization tasks.

primarily target output-level hallucinations or alleviating the textual dominance between image and text modalities, which are possible solutions to cross-modal AD. Representative approaches include attention re-weighting (He et al., 2025; Liu et al., 2024a), distributional contrast or logit control (An et al., 2025; Huang et al., 2024), and layer-wise alignment with visual evidence (Chuang et al., 2023). Concurrently, to ensure accurate spatial reasoning, recent studies (Chen et al., 2025; Cao et al., 2025; Guo et al., 2025; Yang et al., 2025b) increasingly analyze image attention maps to correct misplaced visual focus, mitigating attention drift caused by complex visual environments. While effective for standard VQA, these methods transfer poorly to retrieval-augmented KB-VQA (An et al., 2026), as they overlook the unique challenges introduced by long retrieved contexts, where attention competition and cross-modal interference become more severe. Moreover, they fail to account for the intra-image AD. In contrast, we show that directly addressing these two aspects of AD is fundamental for stabilizing multimodal reasoning and yields larger performance gains.

## 3. Preliminaries

We formalize knowledge-intensive visual question answering (VQA) under both closed-book and retrieval-augmented inference and introduce the attention-based notation used throughout this paper.

**Notation.** We denote image tokens as $\mathbf{I} = \{i_1, \ldots, i_V\}$, where $V$ is the number of image tokens, question tokens as $\mathbf{Q} = \{q_1, \ldots, q_T\}$, where $T$ is the number of question tokens, and retrieved context tokens as $\mathbf{C} = \{c_1, \ldots, c_C\}$, where $C$ is the number of context tokens. During decoding, the model attends to all preceding tokens. We denote the attention weights as $\mathbf{A}(\cdot, \cdot)$, where $\mathbf{A}(\mathbf{X}, \mathbf{Y})$ represents attention weight from query tokens $\mathbf{X}$ to key tokens $\mathbf{Y}$. For decoding step $t$, let $a_{t-1,j}$ denote the attention weight assigned from the $(t-1)$-th decoding token to the $j$-th preceding token. We then define $\rho_t^{(\mathbf{I})} = \sum_{j \in \mathbf{I}} a_{t-1,j}$ and $\rho_t^{(\mathbf{C})} = \sum_{j \in \mathbf{C}} a_{t-1,j}$ as the attention weights allocated to visual tokens and retrieved context, respectively, thereby quantifying the distribution of attention allocated to visual tokens versus retrieved context.

**Inference Paradigms of Closed-book and RAG.** We consider two common inference paradigms that differ in input construction and attention dynamics. In the closed-book setting, no external context is available ($\mathbf{C} = \emptyset$), and the input sequence is $\mathbf{X}_{\text{closed-book}} = [\mathbf{I}, \mathbf{Q}]$, such that reasoning relies solely on the model's parametric knowledge and visual evidence, with attention typically dominated by image tokens for visually grounded questions. In contrast, retrieval-augmented generation (RAG) provides additional external

context, constructing the input as $\mathbf{X}_{\text{RAG}} = [\mathbf{I}, \mathbf{Q}, \mathbf{C}]$ [2] (see Section A for prompts). While this formulation enables knowledge-aware reasoning, it also introduces competition between retrieved context and visual tokens for attention.

**Autoregressive Decoding.** Autoregressive LVLMs adopt causal self-attention, which enforces a strictly left-to-right information flow via a causal mask (Vaswani et al., 2017; Liu et al., 2023; Bai et al., 2025). Given an input sequence $\mathbf{X}$ with query, key, and value representations $\mathbf{Q}$, $\mathbf{K}$, and $\mathbf{V}$ with dimension $d_k$, the masked self-attention at each layer is computed as

$$\text{Attention}(\mathbf{Q}, \mathbf{K}, \mathbf{V}) = \underbrace{\text{softmax}\left(\frac{\mathbf{Q}\mathbf{K}^\top}{\sqrt{d_k}} + \mathbf{M}\right)}_{\text{Attention weights } \mathbf{A}} \mathbf{V},$$

where $\mathbf{M} \in \mathbb{R}^{L \times L}$ is the causal mask defined as

$$\mathbf{M}_{i,j} = \begin{cases} 0, & j \leq i, \\ -\infty, & j > i. \end{cases} \quad (1)$$

This formulation ensures that the query at position $i$ attends only to tokens at positions $j \leq i$, thereby preventing any access to future tokens.

# 4. Attention Distraction (AD) in Retrieval-augmented LVLMs

Although retrieval augmentation improves average performance, Fig. 2 shows that it consistently degrades a non-trivial subset of instances across different LVLMs, suggesting that retrieval alters internal inference dynamics rather than merely providing additional information. To understand the underlying cause, we analyze how retrieval reshapes attention behavior during multimodal generation and identify two forms of attention distraction:

**Cross-modal Attention Distraction.** Across the entire dataset, we observe a systematic redistribution of attention upon the introduction of retrieval. To disentangle the influence of retrieval quality, we utilize high-quality chunks containing relevant information for our analysis. As shown in Fig. 3, retrieval-augmented LVLMs exhibit a 12.1%–41.1% decrease in image-tokens' attention weights, while attention allocated to the textual modality increases significantly. This demonstrates that the retrieved context does not merely provide auxiliary information, but actively reshapes the model's global attention distribution, diverting focus away from visual-modality to text-modality through a cross-modal suppression effect.

**Intra-image Attention Distraction.** Beyond the global reduction of visual modality, retrieval augmentation induces a

---

[2]We omit instruction tokens $\mathbf{S}$ here for simplicity.

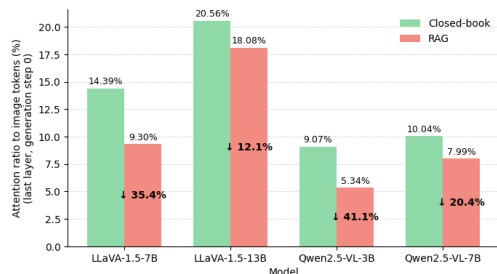

*Figure 3.* **Average image token attention ratio under closed-book and RAG settings across different VLM model families on OK-VQA dataset.** Introducing retrieved textual context consistently reduces the image attention ratio, indicating a distraction of attention from visual inputs toward retrieved context. The attention is extracted at the last layer.

spatial misalignment at the image token level. After removing sink-token effects (Kang et al., 2025; Luo et al., 2025) (details in Appendix C), we compare attention heatmaps generated from visual-important heads (followed (Kang et al., 2025)) at the generation step under closed-book and RAG settings (Fig. 1). In the closed-book setting, attention is concentrated on salient, question-relevant visual tokens, whereas under RAG, the attention shifts toward background or question irrelevant regions.

Together, these results reveal a dual-faceted AD, in which retrieval suppresses global visual attention while simultaneously disrupting focus on the image by shifting attention from salient regions to irrelevant or background regions. This distraction crowds out essential visual evidence and weakens visual grounding, motivating **MAD-RAG**, a training-free intervention that restores robust attention dynamics in retrieval-augmented generation.

# 5. Method

In this section, we present **MAD-RAG**, a training-free inference-time attention intervention that mitigates attention distraction on RAG-based KB-VQA tasks.

### 5.1. Input Construction with Dual-question Format

Under the commonly used RAG prompt structure $X = [\mathbf{I}, \mathbf{Q}, \mathbf{C}]$, all context tokens are positioned after the question tokens. As shown in Eq. (1), for any $q_i \in \mathbf{Q}$ at position $i$ and any $c_j \in \mathbf{C}$ at position $j \geq i$, the causal mask enforces $M_{i,j} = -\infty$. Consequently, the question tokens function as the query for visual attention but remain isolated from the retrieved context. This isolation creates an information bottleneck where the question representation cannot incorporate external knowledge to guide the visual grounding process. This limitation motivates us to decouple the perception to image and external knowledge via a dual-question formulation.

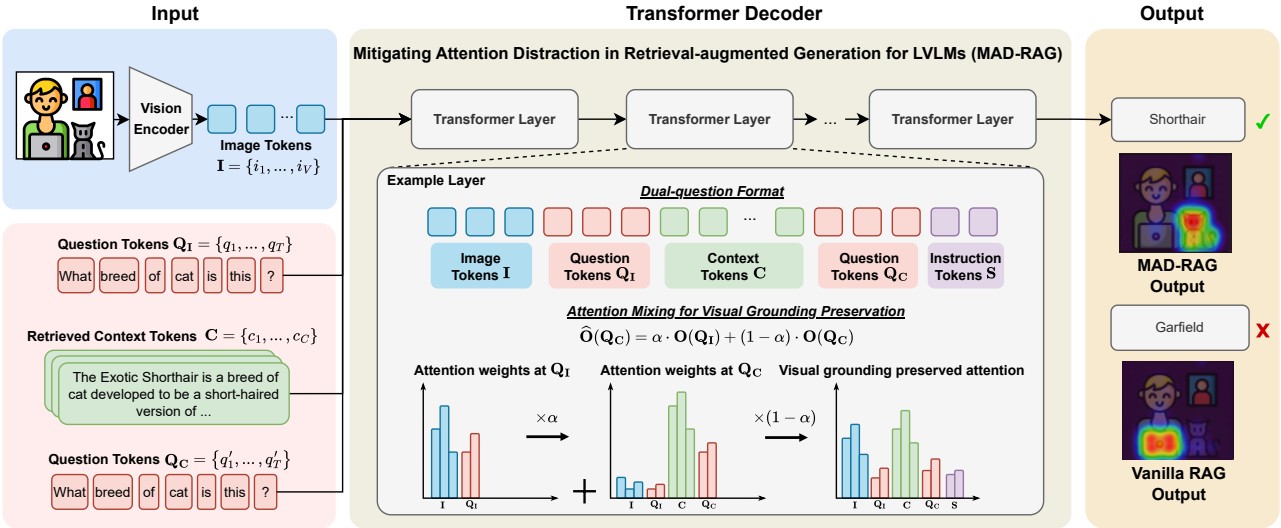

*Figure 4.* **Overview of Mitigating Attention Distraction in Retrieval-augmented Generation for LVLMs (MAD-RAG).** The image is encoded into visual tokens and jointly processed with question and retrieved context tokens by LVLMs. At inference time, attention distributions are intervened by mixing attentions between two question tokens, mitigating context-induced attention distraction. Compared to vanilla RAG, the proposed intervention redirects attention toward question-relevant visual evidence and leads to the correct prediction.

Specifically, to disentangle visual grounding from context integration, we duplicate the question and construct two identical question token groups: (1) **Image-question tokens $\mathbf{Q_I}$:** appended immediately after image tokens. (2) **Context-question tokens $\mathbf{Q_C}$:** appended after retrieved documents. The final input sequence is:

$$\mathbf{X}_{\text{MAD-RAG}} = [\mathbf{I}, \mathbf{Q_I}, \mathbf{C}, \mathbf{Q_C}]. \qquad (2)$$

This design ensures that $\mathbf{Q_I}$ primarily attends to visual tokens, establishing image-prioritized grounding, while $\mathbf{Q_C}$ focuses on integrating retrieved textual context for knowledge-aware reasoning. By decoupling these two roles, the model forms a reliable visual reference before incorporating external information, thereby reducing cross-modal attention interference during subsequent generation. Moreover, because of token similarity, $\mathbf{Q_I}$ can be implicitly attended by $\mathbf{Q_C}$, which further alleviates attention distraction (see Table 4).

### 5.2. Attention Mixing for Visual Grounding Preservation

Although the dual-question formulation explicitly decouples visual perception from context integration, it is not sufficient to completely prevent attention distraction, which can still emerge during decoding under long retrieved contexts due to the model inherent recency bias (Zhao et al., 2021; Press et al., 2021). As the context $\mathbf{C}$ length increase, the attention mechanism disproportionately allocate more attention to the retrieved context compared to the distant image tokens, thereby distracting visual grounding.

To mitigate this, we introduce an attention mixing mechanism to transfer image-grounded attention from $\mathbf{Q_I}$ to $\mathbf{Q_C}$ at each layer. Let $\mathbf{A_{Q_I,I}} = \mathbf{A}(\mathbf{Q_I}, \mathbf{I}) \in \mathbb{R}^{H \times T \times V}$ denote the multi-head attention weights (after softmax) from the image-question tokens $\mathbf{Q}_I$ to image tokens $\mathbf{I}$, and $\mathbf{A_{Q_C,\text{All}}} = \mathbf{A}(\mathbf{Q_C}, [\mathbf{I}, \mathbf{Q_I}, \mathbf{C}]) \in \mathbb{R}^{H \times T \times (V+T+C)}$ denote the attention weights from the context-question tokens $\mathbf{Q_C}$ to all preceding tokens. Here, $H$ is the number of heads, $T$ is the number of question tokens, and $V$ is the number of image tokens. We then intervene by injecting the image-grounded attention weights induced by $\mathbf{Q_I}$ to the attention weights of $\mathbf{Q_C}$ through a convex combination to get the mixed attention weights $\hat{\mathbf{A}}$:

$$\hat{\mathbf{A}}_{\mathbf{Q_C},\text{All}} = \alpha \cdot [\mathbf{A_{Q_I,I}}, \mathbf{0_{Q_I}}, \mathbf{0_C}] + (1-\alpha) \cdot \mathbf{A_{Q_C,\text{All}}}, \quad (3)$$

where $\alpha \in [0, 1]$, and $\mathbf{0_{Q_I}}$ and $\mathbf{0_C}$ are zero tensors matching the attention dimensions of $\mathbf{Q_I}$ and $\mathbf{C}$, respectively. The resulting attention output feature $\hat{\mathbf{O}}(\mathbf{Q_C})$ is computed as:

$$\hat{\mathbf{O}}(\mathbf{Q_C}) = \hat{\mathbf{A}}(\mathbf{Q_C}, [\mathbf{I}, \mathbf{Q_I}, \mathbf{C}]) \cdot \begin{bmatrix} \mathbf{V_I} \\ \mathbf{V_{Q_I}} \\ \mathbf{V_C} \end{bmatrix}, \qquad (4)$$

where $\mathbf{V_I}$, $\mathbf{V_{Q_I}}$, and $\mathbf{V_C}$ denote the value vectors of the image, image-question, and context tokens, respectively. Equivalently, under the scaled dot-product attention formulation, the mixed attention output can be expressed as:

$$\hat{\mathbf{O}}(\mathbf{Q_C}) = \alpha \cdot \mathbf{O}(\mathbf{Q_I}) + (1-\alpha) \cdot \mathbf{O}(\mathbf{Q_C}), \qquad (5)$$

which can be computed efficiently using the pre-computed attention outputs of $\mathbf{Q_I}$ and $\mathbf{Q_C}$. Thus, the weighted combination preserves visual grounding during context integration,

*Table 1.* **Performance (%) comparison with Oracle Contexts across different LVLM families.** We evaluate **MAD-RAG** method against (I) RAG-oriented methods and (II) VLM hallucination mitigation methods (visual-grounding-based)). **Bold** denotes the best performance and underline denotes the second best. † indicates methods strictly incompatible with Qwen2.5's dynamic visual tokenization, resulting in garbled outputs. ‡ indicates methods that cause textual artifacts (e.g., frequent word truncation) on Qwen2.5 due to misaligned attention suppression logic, although they do not produce completely random tokens.

| Method | LLaVA-1.5-7B | | | LLaVA-1.5-13B | | | Qwen2.5-VL-3B | | | Qwen2.5-VL-7B | | |
|---|---|---|---|---|---|---|---|---|---|---|---|---|
| | OK | E-VQA | Info | OK | E-VQA | Info | OK | E-VQA | Info | OK | E-VQA | Info |
| Closed-book (parametric) | 61.65 | 17.57 | 8.01 | 64.32 | 18.80 | 8.14 | 59.97 | 22.93 | 17.88 | 61.92 | 25.15 | 19.58 |
| Vanilla RAG | 65.46 | 53.89 | 44.01 | 66.76 | 61.23 | 46.33 | 62.24 | 75.63 | 51.44 | 63.80 | 82.21 | 45.71 |
| ***I. RAG-Oriented Methods*** | | | | | | | | | | | | |
| CAD (Shi et al., 2024) | 68.44 | 54.88 | 39.06 | 69.38 | 60.19 | 39.06 | 66.24 | 74.11 | 47.98 | **65.45** | 77.89 | 44.12 |
| AdaCAD (Wang et al., 2025b) | 67.01 | 54.69 | 46.45 | 66.15 | 59.47 | 46.45 | 59.11 | 73.04 | 52.04 | 58.84 | 75.65 | 46.33 |
| ALFAR (An et al., 2026) | 65.32 | 53.20 | 38.70 | 67.66 | 60.36 | 45.11 | 62.63 | 75.97 | 51.44 | 63.55 | 82.40 | 48.09 |
| ***II. VLM Hallucination Mitigation Methods (Visual-Grounding-based)*** | | | | | | | | | | | | |
| DoLA (Chuang et al., 2023) | 65.65 | 51.01 | 43.01 | 66.02 | 58.21 | 44.69 | 62.09 | 72.40 | 52.25 | 63.77 | 75.50 | 47.28 |
| VCD (Leng et al., 2024) | 64.48 | 53.20 | 43.86 | 65.55 | 57.92 | 43.61 | 60.81 | 71.57 | 51.68 | 63.17 | 74.99 | 46.98 |
| OPERA† (Huang et al., 2024) | 66.47 | 48.56 | 43.03 | 67.55 | 57.47 | 44.03 | - | - | - | - | - | - |
| VHR‡ (He et al., 2025) | 61.99 | 53.04 | 38.06 | 64.13 | 58.88 | 40.19 | 34.67 | 63.52 | 28.16 | 42.91 | 58.67 | 24.04 |
| SPIN† (Sarkar et al., 2025) | 61.55 | 48.72 | 40.97 | 65.58 | 57.95 | 45.35 | - | - | - | - | - | - |
| **MAD-RAG** (ours) | **70.22** | **63.09** | **50.19** | **71.32** | **69.07** | **51.59** | **66.44** | **80.11** | **54.06** | 64.91 | **83.31** | **48.43** |

ensuring that context-aware reasoning remains focused on question-relevant image regions.

## 6. Experiment

### 6.1. Experimental Settings

**Datasets.** We evaluate **MAD-RAG** on three knowledge-based VQA (KBVQA) benchmarks: **OK-VQA** (Marino et al., 2019), **E-VQA (Encyclopedic VQA)** (Mensink et al., 2023), and **InfoSeek** (Chen et al., 2023). All three datasets require integrating visual evidence with external knowledge, making them suitable for assessing retrieval-augmented VLMs under different knowledge and context conditions. In particular, they differ in the granularity of required knowledge and the complexity of retrieved contexts, enabling a comprehensive analysis of attention behavior in RAG settings. Detailed dataset descriptions, preprocessing details and evaluation criteria are provided in Appendix F.

**Models**. We evaluate **MAD-RAG** on diverse LVLM families and scales, including LLaVA-1.5 (7B, 13B) (Liu et al., 2023) and Qwen2.5-VL (3B, 7B) (Bai et al., 2025), representing both established open-source models and recent high-performance multimodal structures.

**Implementation Details.** We evaluate all models on the official testing splits of each dataset. To control variables and eliminate the impact of retriever performance on overall results, we primarily use the oracle chunks[3] (Lin et al., 2024).

---

[3]Oracle chunks are not strictly equivalent to ground-truth chunks. A detailed explanation of how oracle chunks are con-

**MAD-RAG** is applied to each layer in the LVLMs. In addition, we conduct experiments with real retrieval systems to validate our findings under practical settings, including an image-to-text retriever based on CLIP-L/14-336 (Radford et al., 2021) . All experiments are conducted on a single NVIDIA A100 GPU. To control randomness and ensure reproducibility, we use **greedy decoding** across all experimental settings with seed 0. We report result of $\alpha = 0.5$.

**Baseline Methods.** We compare against representative training-free inference-time methods, grouped into two categories. (i) RAG-oriented methods, including CAD (Shi et al., 2024), AdaCAD (Wang et al., 2025b), and AL-FAR (An et al., 2026), which aim to calibrate or reweight retrieved textual context before or during generation. (ii) VLM hallucination mitigation methods, including DoLa (Chuang et al., 2023), VCD (Leng et al., 2024), OPERA (Huang et al., 2024), VHR (He et al., 2025), and SPIN (Sarkar et al., 2025), which intervene at decoding time to suppress LVLM hallucinations without additional training. All baselines are tuned or following official recommendations. Further details are provided in Appendix G.

### 6.2. Experimental Results

**Comparison with Baselines.** As shown in Table 1, **MAD-RAG** consistently outperforms vanilla RAG and all compared baselines across OK-VQA, E-VQA, and InfoSeek for all evaluated LVLM families. It achieves clear absolute gains over vanilla RAG, with improvements of up to

---

structed is provided in Appendix F.

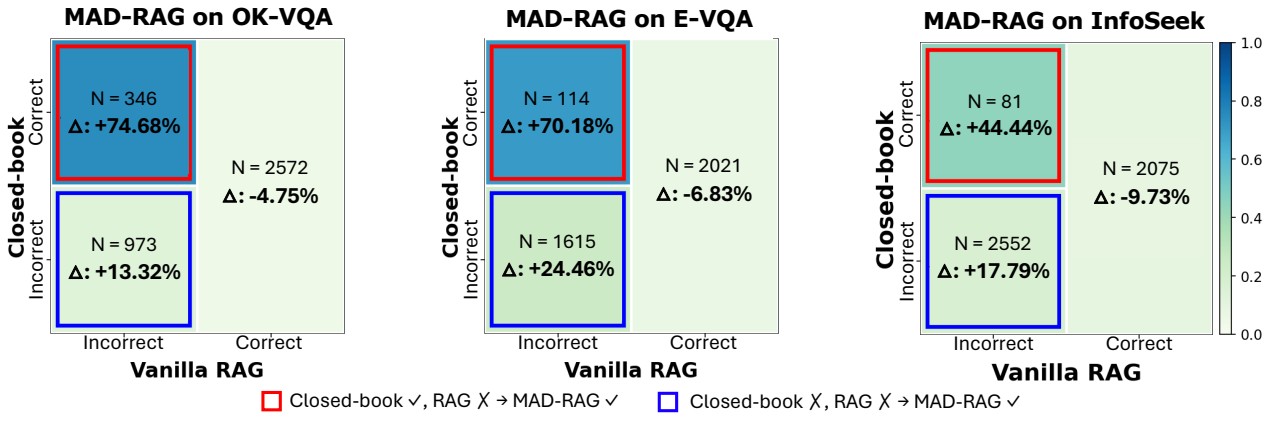

*Figure 5.* **Improvement on different cases with LLaVA-1.5-7B.** Rows and columns indicate whether the instance was correctly or incorrectly by the Closed-book and Vanilla RAG baselines, respectively. Cells report sample size (N), **MAD-RAG**'s accuracy, and the change values (Δ). The red box denotes the attention distraction cases (Closed-book ✓, RAG ×) recovered by **MAD-RAG**.

4.76% on OK-VQA, 9.20% on E-VQA, and 6.18% on InfoSeek. Compared to prior RAG-oriented methods such as CAD, AdaCAD, and ALFAR, **MAD-RAG** achieves mostly consistent improvements, with gains of up to 8.21% across different backbones and datasets, while remaining competitive in the few cases where the best baseline is marginally higher. In contrast, VLM hallucination mitigation methods show limited or even negative gains in RAG settings, where several baselines suffer substantial performance degradation or incompatibility issues.

**Improvement on Failure Mode Cases** To further investigate the effectiveness of **MAD-RAG**, we categorize samples into four quadrants based on the performance of the Closed-book and Vanilla RAG baselines. As illustrated in Fig. 5, **MAD-RAG** significantly rectifies the attention distraction failure mode (Closed-book=1, RAG=0), where the model is misled by retrieved context despite possessing correct internal knowledge. In these cases (red box), we observe remarkable accuracy gains of +74.68%, +70.18%, and +44.44% across OK-VQA, E-VQA, and InfoSeek, respectively. Furthermore, we find that our approach remarkably improves scenarios where both baselines fail (Closed-book=0, RAG=0 as shown in blue box), yielding gains of up to +24.46%. This suggests that **MAD-RAG** effectively uncovers correct answers previously suppressed by both internal biases and retrieval noise, while maintaining stable performance on cases where Vanilla RAG was successful.

**Robustness to Context length.** In this section, we analyze robustness with respect to retrieved context length. As shown in Figure 6, we report performance gains across three datasets with increasing average context lengths, where OK-VQA has the shortest contexts and InfoSeek the longest. Our method consistently achieves positive and the largest gains across all datasets. In contrast, baseline methods such

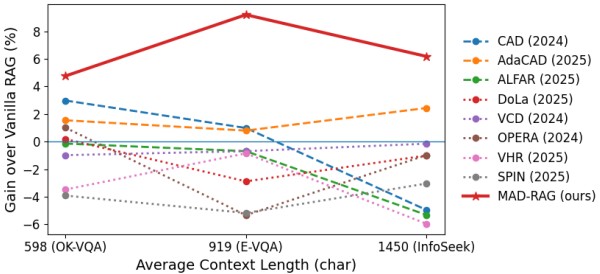

*Figure 6.* Performance comparison of **MAD-RAG** and baseline methods with different retrieved context length.

as ALFAR and CAD exhibit degraded performance as the context length increases, and in some cases even underperform Vanilla RAG.

**Efficiency Comparison.** Table 2 reports the average inference time per sample using LLaVA-1.5-13B across three benchmarks. Compared to the Vanilla RAG baseline, **MAD-RAG** introduces only marginal overhead, with inference time increasing by approximately 9–11% (e.g., from 0.30s to 0.34s on OK-VQA). In contrast, most RAG-oriented calibration and hallucination mitigation methods incur substantially higher latency: CAD and OPERA slow down inference by more than 3×, while AdaCAD, ALFAR, DoLa, VCD, and SPIN consistently introduce 1.3×–2.0× overhead due to their additional decoding-time interventions. Notably, **MAD-RAG** achieves the lowest overhead among all non-trivial baselines, remaining close to real-time performance across all datasets. Overall, the result demonstrate that **MAD-RAG** offers a favorable efficiency–performance trade-off, making it well suited for scalable and latency-sensitive RAG applications.

**Qualitative Analysis.** The qualitative analysis of MAD-

RAG is shown in Appendix B.

### 6.3. Ablation Study

**Robustness to Retrievals.** Table 3 reports results on InfoSeek (LLaVA-1.5-7B) under different retrievers and retrieval quantities. We consider both oracle retrieval and image-to-text retriever using CLIP-L/14-336. **MAD-RAG** consistently outperforms Vanilla RAG regardless of the number of retrieved chunks. Notably, while Vanilla RAG degrades substantially as more noisy contexts are introduced, our approach maintains strong and stable performance, demonstrating robustness to both retrieval quality and quantity.

*Table 3.* **Performance comparison across different retrievers and retrieval quantities. MAD-RAG** consistently outperforms Vanilla RAG regardless of the number of retrieved chunks. We use Infoseek with LLaVA-1.5-7B.

| Retriever | Num. | Accuracy | | |
|---|---|---|---|---|
| | | Closed-book | Vanilla RAG | MAD-RAG (ours) |
| Oracle | 1 | | 43.31 | **50.19** |
| | 3 | | 44.01 | **51.08** |
| | 5 | 8.01 | 40.46 | **52.00** |
| CLIP-L | Top-1 | | 24.21 | **27.32** |
| | Top-3 | | 15.68 | **22.81** |
| | Top-5 | | 9.86 | **21.20** |

**The Attention Injection Weight $\alpha$.** As shown in Fig. 7, **MAD-RAG** consistently outperforms the Vanilla RAG baseline over a broad range of $\alpha$ values from 0.1 to 0.5, indicating that the effectiveness of the proposed attention intervention does not rely on precise hyperparameter tuning. As explained in the Appendix F, oracle chunks are noisier for OK-VQA, favoring larger $\alpha$, while for E-VQA and InfoSeek they are more relevant and often include ground-truth, leading to optimal performance at smaller $\alpha$. In practice,

*Table 2.* **Inference time comparison with LLaVA-1.5-13B (s/sample).**

| Method | OKVQA | E-VQA | Infoseek |
|---|---|---|---|
| Vanilla RAG | 0.30 (1.00x) | 0.29 (1.00x) | 0.31 (1.00x) |
| CAD (2024) | 0.94 (3.12x) | 1.29 (4.45x) | 1.26 (4.05x) |
| AdaCAD (2025) | 0.49 (1.62x) | 0.57 (1.95x) | 0.51 (1.66x) |
| ALFAR (2025) | 0.41 (1.36x) | 0.40 (1.38x) | 0.44 (1.41x) |
| DOLA (2023) | 0.42 (1.39x) | 0.53 (1.81x) | 0.47 (1.52x) |
| VCD (2024) | 0.48 (1.59x) | 0.46 (1.59x) | 0.49 (1.59x) |
| OPERA (2024) | 0.92 (3.06x) | 0.92 (3.18x) | 1.04 (3.37x) |
| VHR (2025) | 0.35 (1.15x) | 0.37 (1.27x) | 0.38 (1.22x) |
| SPIN (2025) | 0.47 (1.55x) | 0.58 (1.97x) | 0.56 (1.78x) |
| **MAD-RAG (ours)** | **0.34 (1.10x)** | **0.33 (1.11x)** | **0.34 (1.09x)** |

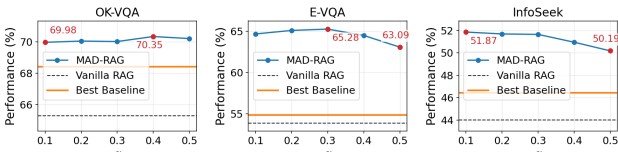

*Figure 7.* **Sensitivity analysis of the attention mixing weight $\alpha$ with LLaVA-1.5-7B.** The performance of **MAD-RAG** remains consistently superior to the Vanilla RAG baseline across a wide range of $\alpha$ values (0.1 to 0.5) on OK-VQA, E-VQA, and InfoSeek. This robustness demonstrates that our method achieves stable visual grounding without requiring exhaustive hyperparameter tuning.

we recommend $\alpha = 0.5$ as a robust default. Overall, **MAD-RAG** remains insensitive to moderate changes in $\alpha$.

**Impact of Question Location.** We conduct an ablation study to examine the effect of question placement in RAG as shown in Table 4 (Swap Q/C). The standard formulation concatenates image, question, and retrieved context as $X_{\text{RAG}} = [\mathbf{I}, \mathbf{Q}, \mathbf{C}]$. We evaluate a variant that swaps the order of the question and context, i.e., $X = [\mathbf{I}, \mathbf{C}, \mathbf{Q}]$, while keeping all other components unchanged. The result reveals that simply placing the question after the context (Swap Q/C) is highly unstable across different backbones, as evidenced by the 1.88% regression in Qwen2.5-VL-7B compared to the vanilla RAG.

**Impact of Attention Mixing.** We also consider an alternative input $X = [\mathbf{I}, \mathbf{Q_I}, \mathbf{C}, \mathbf{Q_C}]$ without attention intervention, serving as a control to separate the effect of question location from architectural changes. This study isolates the impact of attention mixing beyond simple prompt engineering. While duplicating the question (w/o Int.) provides improvement to some extent, **MAD-RAG** achieves the best performance by explicitly intervening in the attention mechanism to ensure the most robust visual grounding.

*Table 4.* **Ablation on question placement on OKVQA (%) with LLaVA-1.5-7B and Qwen2.5-VL-7B.**

| Setting | Input | LLaVA | Qwen |
|---|---|---|---|
| Vanilla RAG | $[\mathbf{I}, \mathbf{Q}, \mathbf{C}]$ | 65.46 | 63.80 |
| Swap Q/C | $[\mathbf{I}, \mathbf{C}, \mathbf{Q}]$ | 68.17 | 61.92 |
| w/o Int. | $[\mathbf{I}, \mathbf{Q_I}, \mathbf{C}, \mathbf{Q_C}]$ | 69.65 | 63.85 |
| **MAD-RAG (ours)** | $[\mathbf{I}, \mathbf{Q_I}, \mathbf{C}, \mathbf{Q_C}]$ | **70.22** | **64.91** |

### 7. Conclusion

We identify Attention Distraction (AD) as a systematic failure mode in retrieval-augmented LVLMs, where retrieved context disrupts visual grounding and causes errors even when parametric knowledge suffices. AD arises from both cross-modal suppression of visual attention and intra-image attention drift. To address this, we propose **MAD-RAG,**

a training-free and plug-and-play method that combines a dual-question formulation with attention mixing to restore focus on question-relevant visual evidence. Extensive experiments show that **MAD-RAG** consistently outperforms prior methods, correcting up to 74.68% of RAG-induced failures with negligible overhead. Future work includes adaptive selection of the mixing weight $\alpha$, deeper theoretical analysis of why retrieved text systematically induces attention distraction, and extending **MAD-RAG** to closed-source models via prompt- or output-level approximations of attention mixing.

## Acknowledgment

We gratefully acknowledge the financial and computational support. The work is supported, in part, by Natural Sciences and Engineering Research Council of Canada (NSERC) through the Discovery Grant (RGPIN-2022-05316) and Alliance Grant (ALLRP 602633-24); Mitacs (IT34007), the IITP grants (RS-2024-00445087, RS-2025-25464461). Furthermore, we thank the Gemini Academic Program, UBC Advanced Research Computing and the Digital Research Alliance of Canada for providing invaluable computational resources.

## Impact Statement

This paper presents work whose goal is to advance the field of Machine Learning. There are many potential societal consequences of our work, none which we feel must be specifically highlighted here.

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

## A. Prompts

The prompt used in this study follows the previous work (Lin et al., 2024; Caffagni et al., 2025).

---

**Closed-book Prompt: OK-VQA**

`{question}`
Answer using a single word or phrase.

---

**Closed-book Prompt: E-VQA & InfoSeek**

Answer the question based on the image.
Question: `{question}`
Do not generate anything but the short answer.
Short answer:

---

**Vanilla RAG Prompt: OK-VQA**

`{question}`
Context:
`{retrieved context}`
Answer using a single word or phrase.

---

**Vanilla RAG Prompt: E-VQA & InfoSeek**

Given the context, answer the question based on the image.
Question: `{question}`
Context:
`{retrieved context}`
If the context does not help with the question, try to answer it anyway. Do not generate anything but the short answer.
Short answer:

---

## B. Qualitative Analysis

As shown in Figure 8, MAD-RAG redirects attention toward question-relevant visual objects, resulting in more focused and semantically aligned attention maps. This provides direct evidence that MAD-RAG mitigates attention distraction and improves visual grounding, explaining the performance gains observed in Table 1.

## C. Difference to Visual Sink Tokens

In Fig. 9, we distinguish our approach from the visual sink tokens identified in (Luo et al., 2025) (originated from the visual encoder, red boxes) and (Kang et al., 2025) (emerging within the LLM, blue boxes). A "sink token" is defined as a token that attracts disproportionately high attention weights despite lacking task-relevant semantic content (Sun et al., 2024; Luo et al., 2025). While previous literature suggests these tokens capture global summaries (Darcet et al., 2024), they often introduce noise in tasks requiring fine-grained, localized visual reasoning (Luo et al., 2025).

In this study, Knowledge-Based VQA (KB-VQA) task focuses local information in images. Furthermore, empirical evidence suggests that visual sink patterns remain invariant between closed-book and RAG settings, obscuring the nuanced differences between these two settings. By filtering and zero-masking these sink tokens, we are able to expose the attention distraction phenomenon and more accurately evaluate the distinct behaviors of each setting.

## D. Context Tokens with High Attention Scores and Corresponding Attention Heatmaps

While our primary contribution lies in identifying the *attention distraction* phenomenon and proposing a simple yet efficient mitigation strategy, we remain intrigued by the mechanism caused this phenomenon. To this end, we extracted the top-3 context tokens exhibiting the highest attention scores. Empirically, we observe that these tokens are predominantly functional words, numerical digits, or semantically irrelevant terms. We further visualized the cross-modal attention heatmaps corresponding to these specific tokens at the inference stage. As illustrated in Fig. 10, the visual attention distribution for these tokens tends to be either uniform or aligned with the distracted regions. These empirical findings

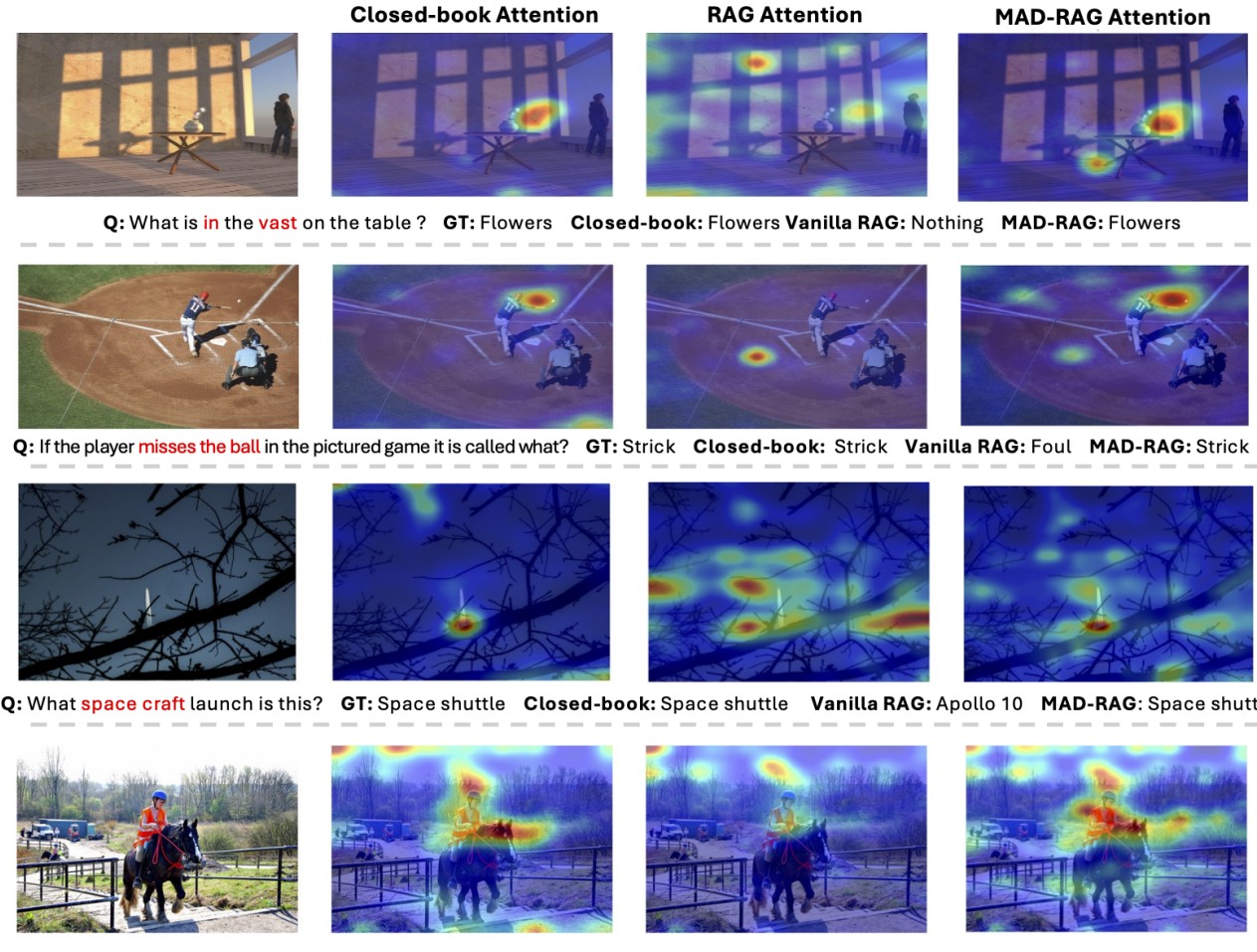

*Figure 8.* **Attention heatmap comparison on OKVQA examples.** Vanilla RAG distract attention away from question-relevant regions, leading to incorrect answers. MAD-RAG restores attention patterns, correctly grounding on the question-related objects and generating correct answers.

identify the potential correlation between specific token types and attention distraction, pointing towards a systematic theoretical analysis for future research.

## E. More Attention Distraction Visualizations

In Fig. 11, we show more visualizations of the attention distraction phenomenon, the important context tokens and the corresponding heatmaps.

## F. Datasets

We evaluate our method on three prominent Knowledge-Based Visual Question Answering (KB-VQA) benchmarks: OK-VQA (Marino et al., 2019), E-VQA (Mensink et al., 2023), and InfoSeek (Chen et al., 2023). The statistical information is shown in Table 5. All of the oracle chunks are from the M2KR benchmark (Lin et al., 2024).

**OK-VQA.** targets open-ended questions that require external knowledge beyond visual recognition. A passage from the knowledge corpus (GS112K) is heuristically labeled as a positive ("oracle" in our paper) pseudo-ground truth if it explicitly contains the ground-truth answer string. Consequently, the retrieval supervision for OKVQA is inherently noisy compared to the other datasets. For this dataset, we select one chunk from the oracle chunks (all baselines use the same chunk).

**E-VQA.** focuses on large-scale encyclopedic knowledge retrieval. Unlike OK-VQA, the oracle chunks in E-VQA are

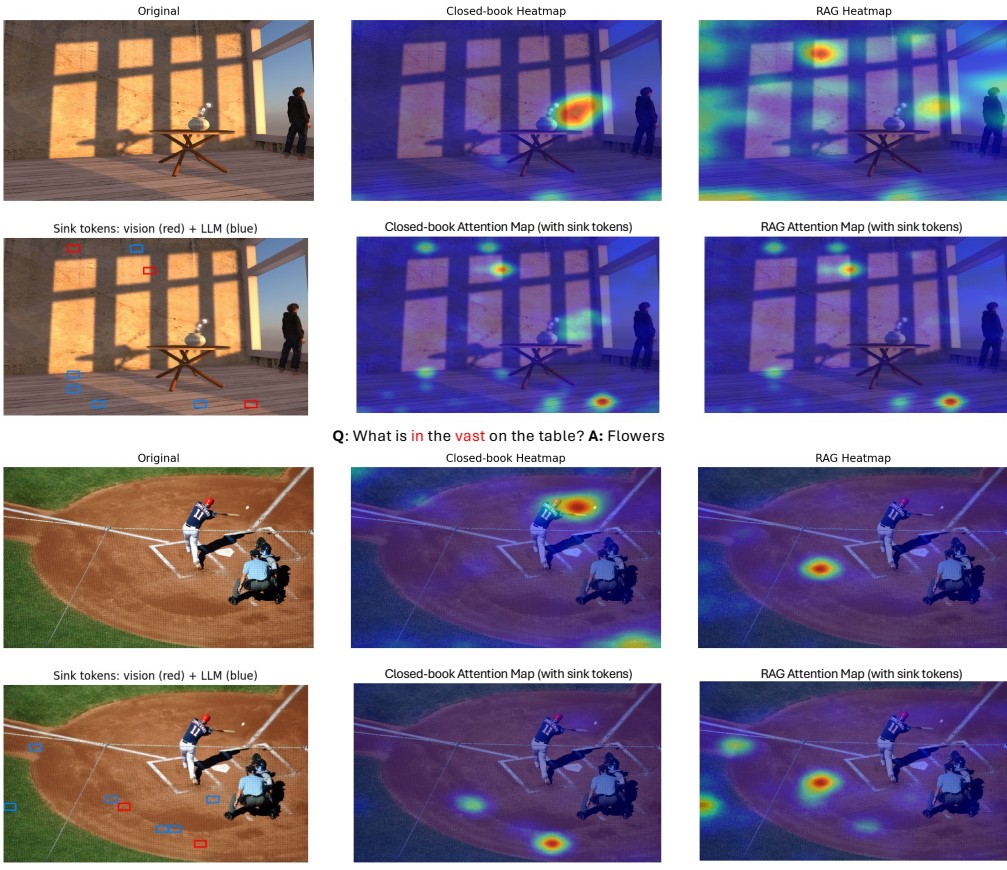

*Figure 9.* The comparison of attention heatmaps without/with visual sink tokens. If the visual sink tokens are not filtered out, the high attention areas are identical to sink tokens, hindering the analysis of attention distraction.

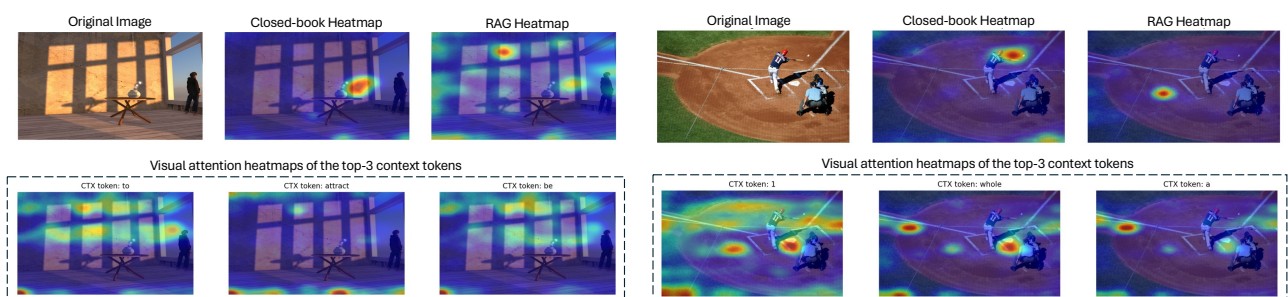

*Figure 10.* Visual attention heatmaps of the most important context tokens

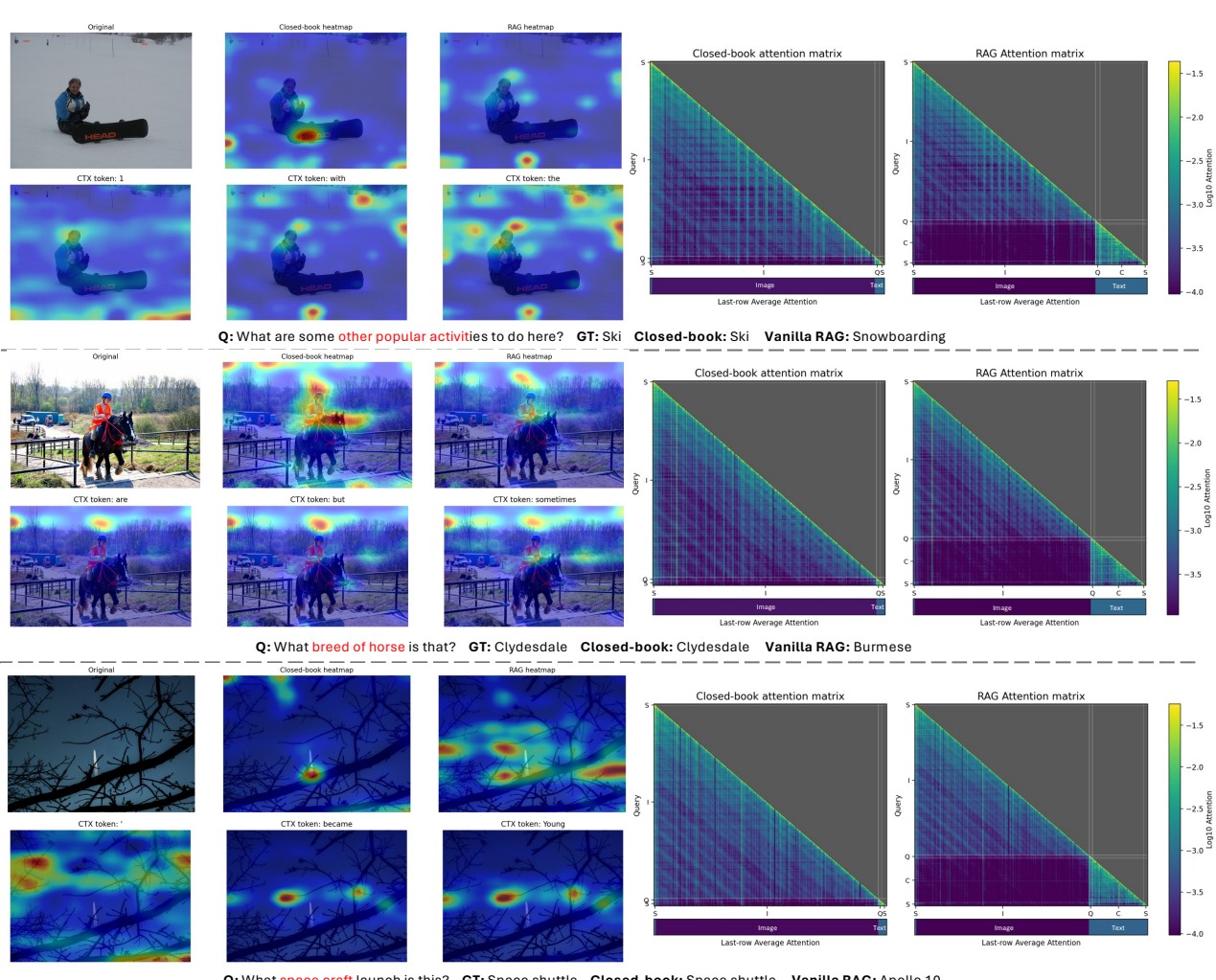

*Figure 11.* More attention distraction visualizations.

constructively paired. The dataset generation process involved formulating questions based on specific textual snippets; thus, the source passage used to generate a given question is deterministically assigned as the ground-truth evidence, ensuring a noise-free mapping between the query and the knowledge chunk. For this dataset, the chunks are concatenated together in the original dataset, we directly use the whole text.

**InfoSeek.** is designed for fine-grained visual information seeking, requiring models to link visual entities to specific informational descriptions. The oracle chunks are established through human-verified annotation. Each image-question pair is explicitly linked to a specific Wikipedia section or summary that describes the visual entity in question, providing high-quality golden standard supervision for the retrieval task. For this dataset, we use the first three chunks and concatenate them together with the format "`[1] chunk 1 \n [2] chunk 2 \n [3] chunk 3`".

*Table 5.* Statistics of OKVQA, E-VQA, and InfoSeek Datasets.

| Dataset | Test Examples | Knowledge Base (Passages) |
| --- | --- | --- |
| OK-VQA | 5,046 | 110,000 |
| E-VQA | 3,750 | 50,000 |
| InfoSeek | 4,708 | 100,000 |

## G. Baselines

**CAD** (Shi et al., 2024) is a training-free inference-time decoding strategy that effectively mitigates hallucinations driven by prior knowledge by contrasting output distributions with and without context, thereby significantly enhancing the faithfulness of generation to the retrieved context.

**ADACAD** (Wang et al., 2025b) introduces an adaptive decoding strategy that dynamically calibrates the reliance on retrieved context by measuring the Jensen-Shannon divergence between parametric and context-aware distributions, thereby effectively handling varying degrees of knowledge conflict without over-correction.

**ALFAR** (An et al., 2026) observes the attention bias on image is larger than answer-related context tokens. It proposes a training-free, plug-and-play framework that mitigates attention bias and knowledge conflicts in multimodal RAG by dynamically reallocating attention from visual to context tokens and adaptively fusing parametric and retrieved knowledge logits.

**DoLa** observes that factual knowledge tends to localize in higher transformer layers. It improves generation faithfulness via a contrastive decoding strategy that amplifies factual signals by maximizing the logit difference between the final layer and dynamic intermediate layers.

**VCD** (Leng et al., 2024) introduces a training-free visual contrastive decoding strategy that mitigates object hallucinations in Large Vision-Language Models by contrasting output distributions from original and distorted visual inputs, effectively calibrating the model's over-reliance on statistical biases and language priors.

**OPERA** (Huang et al., 2024) is a training-free decoding method that mitigates hallucinations by penalizing the model's "over-trust" in specific summary tokens through a logit penalty and a retrospection-allocation strategy, thereby encouraging better attention to visual content during beam search.

**VHR** (He et al., 2025) identifies that the model's over-reliance on its prior language patterns is closely related to hallucinations. It proposes a a training-free approach to mitigate hallucination by enhancing the role of vision-aware attention heads in certain layers.

**SPIN** (Sarkar et al., 2025) suggests that hallucinations can be attributed to a dynamic subset of attention heads in each layer. Leveraging this insight, for each text query token, they selectively suppress attention heads that exhibit low attention to image tokens, keeping the top-k attention heads intact.

### G.1. Implementations

We largely followed official implementations and standard evaluation protocols on the KB-VQA dataset, utilizing greedy decoding where applicable. We conduct specific adaptations to optimize performance or ensure compatibility of baselines. Specifically, we extended DoLa, VCD and VHR (Chuang et al., 2023; Leng et al., 2024; He et al., 2025) to Qwen2.5-VL (Bai et al., 2025) following the method described in the original papers. We re-tuned hyperparameters for CAD ($\alpha = -0.2$) and AdaCAD ($\alpha = -0.5$) to better suit LVLMs. We adjusted Opera's configuration (reducing beam size/candidates to 2 and window size to 256) to satisfy GPU memory constraints under the setting of RAG.

## H. Generalizability across VLMs

To verify the universality of our approach, we extend **MAD-RAG**s to more VLM structures such as Qwen3-VL (Yang et al., 2025a), InternVL-3.5 (Wang et al., 2025d), Phi3V (Abdin et al., 2024), GLM-4.5V (Hong et al., 2025) and Qwen2.5-VL with larger parameters. As shown in Table 6, **MAD-RAG** consistently improves performance across all tested families, demonstrating that the attention distraction is a fundamental issue in VLM RAG, and our solution is model-agnostic.

*Table 6.* **Universal Effectiveness across Model Families.** Performance comparison across four settings: Closed-book, Vanilla RAG, MAD-RAG, and Ours. **Bold** indicates the best result per model.

| Model | Method | OKVQA | E-VQA | Infoseek |
|---|---|---|---|---|
| Qwen3-VL-4B | Vanilla RAG | 65.96 | 75.49 | 52.12 |
| | MAD-RAG | **67.67** | **83.84** | **53.29** |
| Qwen3-VL-8B | Vanilla RAG | 67.03 | 77.60 | 52.66 |
| | MAD-RAG | **69.19** | **84.03** | **55.93** |
| Qwen2.5-VL-32B | Vanilla RAG | 53.44 | 84.24 | 46.20 |
| | MAD-RAG | **55.20** | **87.33** | **46.84** |
| Qwen2.5-VL-72B | Vanilla RAG | 69.81 | 83.49 | 49.53 |
| | MAD-RAG | **70.70** | **86.13** | **51.30** |
| InternVL-3.5 | Vanilla RAG | 62.30 | 80.96 | 52.80 |
| | MAD-RAG | **63.92** | **84.88** | **54.33** |
| Phi3-Vision | Vanilla RAG | 65.66 | 76.13 | 52.06 |
| | MAD-RAG | **66.76** | **79.81** | **52.91** |
| GLM-4.5V | Vanilla RAG | 58.56 | 85.68 | 53.57 |
| | MAD-RAG | **59.57** | **86.86** | **53.74** |

## I. The Attention Injection Layers.

Prior studies (Tang et al., 2025; Sarkar et al., 2025) have shown that different transformer layers in LVLMs serve distinct functional roles: early layers primarily encode visual features, middle layers facilitate vision–language alignment, and later layers are more responsible for high-level reasoning. In Table 7, we report the performance of applying **MAD-RAG** at different subsets of layers. The results indicate that **MAD-RAG** is robust to the choice of intervention layer and consistently outperforms vanilla RAG across all settings, with the best performance achieved when the intervention is applied to all layers.

*Table 7.* **Performance of MAD-RAG with different intervention layers on OK-VQA (%).**

| Layers | LLaVA-1.5-7B | LLaVA-1.5-13B | Qwen2.5-VL-3B | Qwen2.5-VL-7B |
|---|---|---|---|---|
| Early | 70.23 | 70.93 | 66.15 | 64.64 |
| Middle | 69.87 | 71.37 | 65.47 | 64.24 |
| Later | 70.10 | 71.22 | 65.49 | 64.12 |
| All | 70.22 | 71.32 | 66.44 | 64.91 |

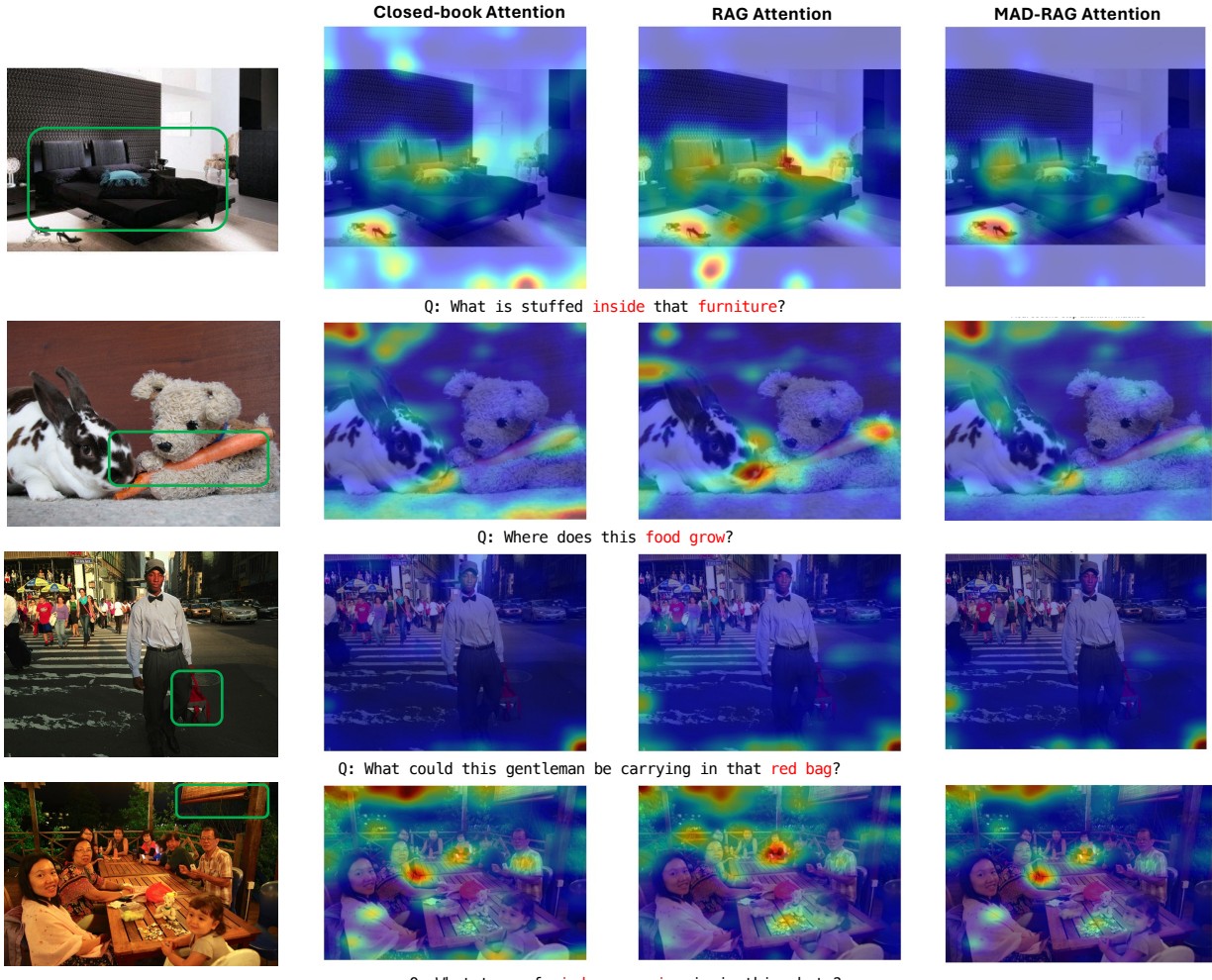

*Figure 12.* **Attention heatmaps for cases where Vanilla RAG is correct but MAD-RAG fails.** In these examples, the Closed-book attention does not focus on question-relevant regions (highlighted in green), indicating the model's inherently weak visual perception. Vanilla RAG succeeds by exploiting language bias in retrieved chunks rather than visual grounding. MAD-RAG brings back the bad closed-book attention, disrupting this text-driven shortcut and leading to incorrect answers.

## J. Boundary Conditions of MAD-RAG

We further analyze the boundary conditions of MAD-RAG, where vanilla RAG produces correct predictions while MAD-RAG fails (RAG ✓, MAD-RAG ✗), as illustrated in Fig. 5. For these instances, we examine the closed-book visual attention maps (Fig. 12), and observe that the model's attention often does not focus on question-relevant objects or regions, indicating inherently weak visual perception for these samples. In such cases, vanilla RAG's correct predictions are typically driven by language bias in the retrieved oracle chunks rather than grounded visual understanding. To verify this, we follow the analysis in Fig. 13 and measure attention scores over the retrieved text with the image removed. We find that the model assigns the highest attention to answer-supporting words or phrases even without visual input, demonstrating that the prediction can be largely driven by textual signals alone. Consequently, when MAD-RAG incorporates closed-book visual attention, it may introduce noise from irrelevant image regions, disrupting this text-based shortcut and leading to incorrect answers. This behavior reveals an important boundary condition of MAD-RAG: its effectiveness relies on the model possessing sufficient visual grounding ability such that meaningful visual attention can be leveraged. When this assumption does not hold, the mixing process can become counterproductive.

While a small number of cases MAD-RAG did not outperform vanilla RAG, as shown in Table 1 and Fig. 5, the number of cases MAD-RAG successfully recovers far exceeds the cases where vanilla RAG introduces errors, leading to substantial

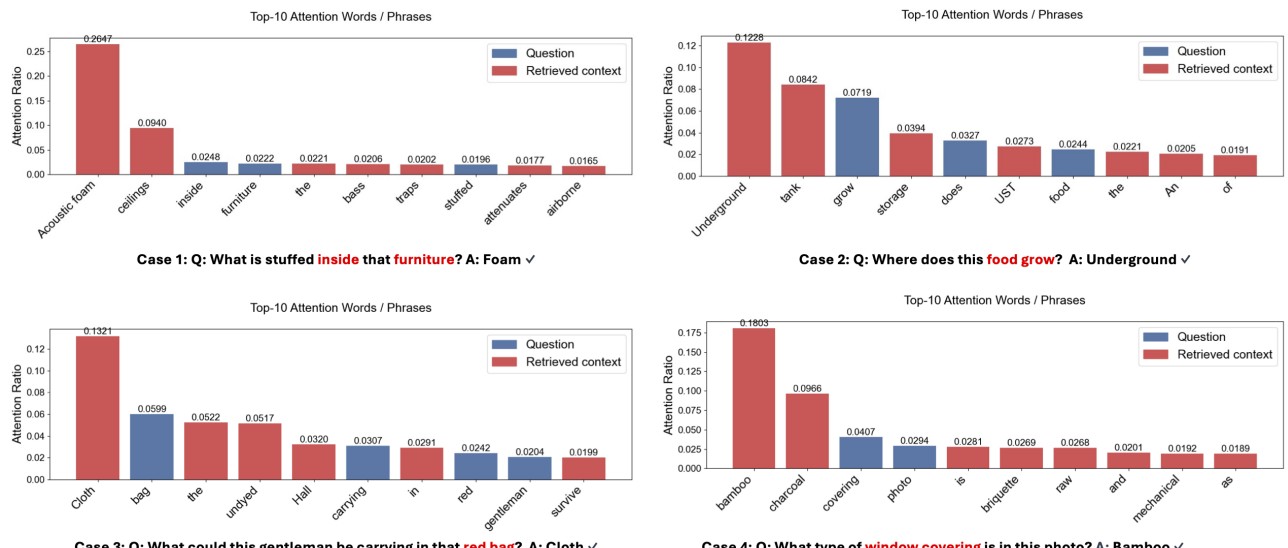

*Figure 13.* **Examples in the image-removed setting with answers and top-10 attention words/phrase.** The model still generates the correct answer even when the image is removed. It also shows that answer-supporting words/phrases receive the highest attention.

overall accuracy gains across all datasets and model families.

