# OpenReview forum: "When RAG Hurts: Diagnosing and Mitigating Attention Distraction in Retrieval-Augmented LVLMs"
_ICML.cc/2026/Conference — ICML 2026 regular_

### Official Review · Reviewer_guYh · 2026-02-25

**Soundness:** 3
**Presentation:** 3
**Significance:** 4
**Originality:** 4
**Overall Recommendation:** 5
**Confidence:** 4

**Summary:**

This paper identifies and addresses a counter-intuitive failure mode in Retrieval-Augmented Generation (RAG) for Large Vision-Language Models (LVLMs), termed "Attention Distraction" (AD). The authors observe that providing external context—even when relevant/correct—can sometimes degrade performance on questions the model could originally answer correctly using its parametric knowledge. Through attention analysis, they diagnose two forms of AD: Cross-Modal AD and Intra-Image AD. To mitigate this, the authors propose MAD-RAG, which uses a dual-question formulation to decouple visual grounding from context integration, combined with an attention mixing strategy to inject image-prioritized attention into the final generation step. Extensive experiments demonstrate that MAD-RAG consistently outperforms vanilla RAG and existing hallucination-mitigation baselines.

**Compliance With Llm Reviewing Policy:**

Affirmed.

**Final Justification:**

The authors have comprehensively and convincingly addressed all the concerns I raised. They performed extensive supplementary experiments on the adaptive α parameter and low-quality retrieval scenarios, which robustly validate the effectiveness and robustness of the MAD-RAG method. Overall, the work is solid and the experimental validation is thorough. I believe it meets ICML’s acceptance criteria and have therefore raised my review score.

**Key Questions For Authors:**

The paper shows that \alpha is stable over a range and recommends a default value. Have you explored adaptive \alpha strategies conditioned on context length, retrieval confidence, or early-layer attention statistics?
If the above questions and weaknesses are resolved, I will improve my score.

**Limitations:**

The authors have adequately discussed limitations, specifically noting the need for adaptive selection of the mixing weight $\alpha$ and the lack of a deep theoretical analysis for why retrieved text systematically induces distraction. They also mention the scope is currently limited to open-source models where attention weights are accessible.

**Strengths And Weaknesses:**

Strengths:

1.	This paper clearly states the motivation and analyzes the challenges of the existing multimodal RAG, which shows that retrieval can actively harm reasoning by disrupting visual attention mechanisms, rather than just introducing noise. Describing this phenomenon as "Attention Distraction" provides a valuable perspective for future research.
2.	The proposed MAD-RAG method is training-free and introduces very low latency, which is a significant advantage over other decoding-time interventions like CAD or OPERA. This makes it highly practical for deployment.
3.	The authors evaluate the method across multiple datasets (OK-VQA, E-VQA, InfoSeek) and different model families (LLaVA-1.5, Qwen2.5-VL, Qwen3-VL), demonstrating consistent improvements.

Weaknesses:

1.	Dynamic $\alpha$ Feasibility: The current method uses a fixed mixing weight $\alpha$. Have you explored making $\alpha$ dynamic, perhaps conditioned on the similarity between the retrieved context and the visual features, or the model's internal uncertainty?
2.	Dependence on Oracle/High-Quality Retrieval for Diagnosis: Much of the diagnostic analysis relies on "Oracle" or high-quality chunks to prove that AD exists, even with good context. While it is necessary to isolate the phenomenon, further discussion on how "Intra-Image Attention Distraction" behaves under adversarial or completely irrelevant retrieval scenarios would strengthen the robustness claims.
3.	The spacing between paragraphs in the paper is inconsistent; you should check it carefully.

---

> ### Author Rebuttal · Authors · 2026-03-29
>
> ----- W1 -----
>
> We thank the reviewer for this insightful suggestion. We have conducted additional experiments to explore adaptive α strategies. We investigate four adaptive strategies:
> 1. **Adaptive α (context length):** Under the oracle chunk setting, we compute the minimum and maximum retrieved context lengths over the dataset and linearly map each sample’s context length to α. This assigns larger α values to longer contexts.
> 2. **Adaptive α (confidence)**: Under the oracle chunk setting, we run MAD-RAG with α ranging from 0.1 to 0.9 in steps of 0.1, and take the answer with the highest probability across the nine predictions as the final output.
> 3. **Adaptive α (majority voting):** Under the oracle chunk setting, we run MAD-RAG with α ranging from 0.1 to 0.9 in steps of 0.1, and take the majority-voted answer across the nine predictions as the final output.
> 4. **Adaptive α (retrieval confidence):** Under the real retriever setting, we construct α by linearly mapping α to (1 - similarity score), so that lower retrieval confidence leads to larger α values.
>
> The results are shown below:
>
> Table: Adaptive α experiments on LLaVA-1.5-7B (Oracle Retrieval)
> | Method | OK-VQA | E-VQA | InfoSeek |
> |---|---|---|---|
> | Fixed α (original manuscript, Table 1) | **70.22** | **63.09** | **50.19** |
> | Adaptive α (context length) | 70.29 | 63.04 | 48.28 |
> | Adaptive α (confidence) | 70.72 | 62.29 | 50.30 |
> | Adaptive α (majority voting) | 70.03 | 65.28 | 51.66 |
>
> Table: Adaptive α with a real retriever CLIP-L on InfoSeek (w/ top-1 chunk)
> | Method | InfoSeek |
> |---|---|
> | Fixed α (original manuscript, Table 3)| **27.32** |
> | Adaptive α (context length) |27.15|
> | Adaptive α (confidence) |27.42|
> | Adaptive α (majority voting) |27.32|
> | Adaptive α (retrieval confidence) | 27.08 |
>
> We observe the following:
> 1. Adaptive α based on context length and retrieval confidence shows little to no improvement over the fixed α, suggesting that these signals with the current strategies do not correlate well with the optimal mixing weight.
> 2. Adaptive α based on confidence and majority voting achieve better results on some datasets, but require 9x higher inference cost, which contradicts our design goal of maintaining negligible computational overhead (as shown in Table 2).
>
> Regarding the reviewer's suggestion of conditioning on early-layer attention statistics: this is an appealing direction, as early-layer image attention ratios could serve as a lightweight proxy for the degree of AD at each sample. We have not yet experimented with this but view it as a promising future direction that could yield a single-pass adaptive strategy without the cost of multi-run approaches. We will add this discussion to the revision.
>
> ----- W2 -----
>
> We appreciate this important question.  As shown in Table 3, we evaluate with real retrievers (CLIP-L Top-5) where retrieved chunks are partially **irrelevant**. MAD-RAG still achieves substantial improvements under these conditions (e.g., InfoSeek: 9.86% → 21.20%). Furthermore, we conduct an experiment with random chunks as the **irrelevant/adversarial retriever** setting on OKVQA (α=0.9):
> | Method | OKVQA (%) |
> |---|---|
> | closed-book | 61.65 |
> | Vanilla RAG (random chunk) | 53.36 |
> | **MAD-RAG** (random chunk) | 60.21 |
>
> Random retrieval causes Vanilla RAG to drop 8.29% below closed-book, demonstrating severe AD even with completely irrelevant content, while MAD-RAG recovers the vast majority of this degradation. The results further confirm that the structural attention distraction we identify persists across different levels of semantic conflict, and our method remains effective even when retrieval quality is low.
>
> ----- W3 -----
>
> We thank the reviewer for pointing this out. We have carefully proofread the formatting and corrected all spacing inconsistencies throughout the manuscript.

---

> > ### Author Rebuttal · Reviewer_guYh · 2026-04-01
> >
> > The authors have provided comprehensive and convincing responses to all my concerns.They have conducted sufficient supplementary experiments on adaptive α and low-quality retrieval, which fully verify the effectiveness and robustness of MAD-RAG. Their responses are basically sufficient, and I have no further objections.

---

> > > ### Author Response · Authors · 2026-04-01
> > >
> > > Thank you again for your positive feedback and for recognizing our contributions. We are very glad to know that our responses have addressed your concerns. We would appreciate if you would consider adjusting your score or let us know if there are any remaining questions we could address. We also thank you for your valuable suggestions, which will help us further enhance the paper.

---

### Official Review · Reviewer_KFM1 · 2026-03-02

**Soundness:** 2
**Presentation:** 3
**Significance:** 2
**Originality:** 3
**Overall Recommendation:** 3
**Confidence:** 3

**Summary:**

This paper investigates the multimodal RAG problem and identify a phenomenon termed Attention Distraction. The authors empirically find that external text contexts suppress visual attention both globally and locally. To address this, the authors propose MAD-RAG, a training-free framework that utilizes a dual-question format and an attention mixing mechanism to re-align the model's focus. Experimental results on KB-VQA benchmarks demonstrate that the proposed method significantly outperforms vanilla RAG by mitigating attention shifts during inference.

**Compliance With Llm Reviewing Policy:**

Affirmed.

**Final Justification:**

The authors' rebuttal partially addresses my concerns.

**Key Questions For Authors:**

- Confusion about the motivation. The observed "Attention Distraction" may be a symptom rather than the root cause. A failure in semantic integration, e.g., being misled by textual distractors, could cause the model to disregard visual tokens. Thus, attention shifts might merely visualize a reasoning error already committed, rather than being the primary driver of the failure. Furthermore, this paper focuses on structural attention competition but overlooks deep semantic contradictions. If retrieved context subtly conflicts with visual evidence, adjusting attention weights provides a mechanical fix that fails to address logic-level misjudgments or resolve the tension between conflicting modalities.
- Concerns regarding necessarily of Attention Mixing. Figure 7 shows that performance remains consistently high and nearly flat across a wide range of λ values. The lack of results for edge cases (λ=0 or λ=1) makes it difficult to discern if the "mixing" process is necessary at all. If λ=0 (relying solely on the Dual-question structure) yields similar results, the proposed attention-mixing theory would lose its empirical justification. The results in Table 4 further increase my concerns, where dual-question format appears to contribute the vast majority of the performance improvement.
- Lack of conceptual novelty. The idea of adjusting the attention weights based on attention on image is not new in the field fro multimodal large language model.

**Limitations:**

The limitation is not discussed.

**Strengths And Weaknesses:**

- This paper explores an interesting question of attention distraction in multimodal RAG framework.
- This paper is well-illustrated and easy to follow.
- The experiments demonstrate the effectiveness of proposed framework.

---

> ### Author Rebuttal · Authors · 2026-03-29
>
> ------ Q1 ------
>
> The reviewer raises two concerns: (a) AD may be a symptom of being misled by textual distractors, and (b) adjusting attention is a mechanical fix that cannot resolve deeper semantic contradictions. We believe our evidence below addresses both concerns:
>
> **AD persists without text distractors.** In lines 86-89, we stated the motivation: RAG degrades a non-trivial subset of instances that LVLMs answer correctly in the closed-book setting **(retrieved context is accurate and relevant)**. We controlled the textual distractors by using the oracle chunks (Sec. 6.1 and Appendix E), yet the performance still degrades significantly (Fig. 2). This shows that AD is a structural phenomenon arising from how autoregressive attention distributes capacity across modalities, even when the retrieved text is distractor-free.
>
> **Attention mixing restores correct answers.** If the model had already committed a semantic error and attention shifts were merely a symptom, forcing re-attention to image tokens should not recover the correct answer. However, MAD-RAG consistently shows strong gains even under imperfect retrieval (textual distractors) in Tab. 3 with up to ~12% gains.
>
> We acknowledge semantic conflicts can co-occur with AD. Our evidence shows AD is an independent failure mode, and addressing it can lead to gains.
>
> ------ Q2 ------
>
> We believe the reviewer is referring to the attention mixing weight α in Fig. 7, rather than λ. To avoid confusion, we use
> α in the following response. First, we wanted to clarify that we have already included α=0 results in Tab. 4, row “w/o Intervention” (i.e., dual-question format only, no attention mixing).  For both Qwen and LLaVA models, MAD-RAG (α$\neq$0) outperforms “w/o Intervention” (α=0). Also, in the Qwen experiment, α=0 only marginally improves over Vanilla RAG (+0.05%), showing that attention mixing is helpful (+1.06%).
> As discussed in Sec. 5.2 (lines 256–260), the dual-question formulation decouples visual grounding from context integration but cannot fully prevent AD under long retrieved contexts due to the model's inherent recency bias. Attention mixing addresses this complementary aspect by explicitly transferring image-grounded attention from $Q_I$ to $Q_C$ at each layer.
>
> We did not provide the result of α=1 since knowledge-based VQA inherently depends on both visual evidence and textual knowledge, α should naturally fall in (0,1) to mix the two sources. We provide the result of α=1 (LLaVA-1.5-7B): OK-VQA-68.91%, E-VQA-51.76%, InfoSeek-45.48%, which shows lower performance than MAD-RAG (α$\in$ (0,1)) as expected. We will add the boundary result of α=0 and 1 to Fig. 7 in the revision.
>
> Second, the consistent results when α varies from 0.1 to 0.5 can be viewed as a practical strength, indicating MAD-RAG is robust to hyperparameter and does not need exhaustive tuning. This is a desirable property for a training-free method.
>
> ------ Q3 ------
>
> We appreciate the opportunity to reiterate how our contributions differ from prior attention-based methods. MAD-RAG's novelty lies in the diagnostic finding that motivates it and the structurally distinct intervention it introduces.
>
> **Existing methods operate under the opposite assumption**. Prior attention-based methods (VCD, OPERA, VHR, SPIN) assume over-reliance on language priors and boost visual attention, not particularly for the RAG setting, where additional text content is added into the VQA decoding process. The most relevant work, ALFAR (2025) claims VLMs themselves do not provide sufficient attention to retrieved text in RAG and reallocates more attention toward retrieved text in RAG. Our work reveals a different diagnosis of attention by **analyzing the image token attention shift before and after RAG**: retrieved text (even correct) **distracts** visual attention, causing failures on questions answerable without retrieval. As Table 1 shows, all these methods yield limited or negative gains in RAG. This confirms that AD is a genuinely new finding that existing methods fail to capture.
>
> **MAD-RAG is structurally distinct from attention reweighting**. Unlike prior methods that reallocate attention on different modalities, MAD-RAG introduces a dual-question decoupling strategy that separates visual grounding from context integration (Eq. 2), then combines with attention mixing (Eq. 3–5) to preserve image-conditioned evidence. This is a structured intervention to address both cross-modal and intra-image distraction, not simple attention reweighting. No prior work introduces this decoupling strategy.
>
> **Practical advantages:** MAD-RAG is training-free, no retriever modification, negligible overhead, consistently outperforming all baselines across multiple models and datasets.
>
> We are glad to see other reviewers recognize our work as “provides a valuable perspective for future research” (guYh), “highly important” (yHK6), and “novel” (rqPV). We hope our justification could help you reconsider our contribution.

---

> > ### Author Rebuttal · Reviewer_KFM1 · 2026-04-01
> >
> > - Thanks for the clarification. I get the core contributions of your work, but the research scope is quite limited. It only focuses on scenarios with oracle chunks and fails to explore Attention Distraction in more realistic non-gt chunk scenarios (e.g., noisy retrieval), which hurts the generalizability of your conclusions.
> > - Thanks for clarifying the data, but I’m more confused now. Figure 7 shows alpha is insensitive between 0.1–0.5, but is there a sharp performance jump between alpha=0.0 and 0.1? If yes, why? If not, how does it align with your claim that attention mixing helps significantly?
> > - Your explanation isn’t convincing enough, and my evaluation stands: adapting methods like VCD to the RAG setting doesn’t meet ICML’s contribution standards. This is just scenario adaptation, not methodological innovation, which contradicts your claim of “structural distinctiveness.”

---

> > > ### Author Response · Authors · 2026-04-04
> > >
> > > Thank you for the detailed follow-up. We appreciate the opportunity to clarify several points regarding the remaining concerns.
> > >
> > > **1. On evaluation under realistic (non-oracle) retrieval.**
> > >
> > > Thanks for raising the concern. We clarify that MAD-RAG is robust beyond oracle scenarios. While oracle chunks were used initially to disentangle retrieval from generation, our method generalizes effectively to realistic and noisy settings. To further support this, we highlight:
> > > - **Performance with real retrievers**: Under a **realistic CLIP-L retriever** (**Table 3**), MAD-RAG consistently recovers attention failures and yields up to ~12% improvement.
> > > - **Robustness in adversarial settings**: (As detailed in our response to **Reviewer guYh**), when introducing random misleading chunks, Vanilla RAG suffers an 8.29% drop, whereas MAD-RAG successfully recovers a significant portion of this degradation.
> > >
> > > These results explicitly address the concern regarding non-gt chunk scenarios and demonstrate our method's broad generalizability.
> > >
> > > **2. On α=0 vs. α>0.**
> > >
> > > Thank you for raising this point, which allows us to clarify the settings of α=0 and α>0. **Any α>0 activates the attention mixing mechanism**, in contrast, the **attention mixing is deactivated in α=0 setting**, where only the dual-question formulation remains. To answer your question directly: Yes, there is a performance jump from α=0 to α=0.1, since attention mixing is built on the top of the dual-Question format. We would like to clarify our evaluation below:
> > >
> > > - **The Foundation (Dual-Question, α=0):** This formulation provides the initial performance gain by decoupling the roles of different text tokens at the prompt level. It explains why α=0 already outperforms vanilla RAG.
> > > - **The Further Improvement (Attention Mixing, α>0):** The dual-question format does not explicitly intervene in the cross-modal attention flow. By activating Attention Mixing (any α>0), we explicitly recover image-conditioned evidence that is otherwise suppressed by retrieved distractors. This is why we observe a distinct performance jump from α=0 to α=0.1.
> > > - **Robustness via Insensitivity:** Once this "further improvement" is activated, the exact value of α becomes insensitive within a large value range. We view this as a practical strength, proving that our method provides a reliable "boost" without requiring exhaustive hyperparameter search.
> > >
> > > In conclusion, the dual-question format provides the necessary **foundation**, and the attention mixing leads to the final **performance peak**.
> > >
> > > **3. On novelty and relation to prior work.**
> > >
> > > We appreciate the opportunity to further highlight our contribution and novelty over the prior work. We agree that, **on the surface, MAD-RAG shares the training-free, attention-intervention philosophy of prior methods.** However, we do not view this as a limitation or a simple scenario adaptation. Prior methods operate at the **modality level** (suppressing language vs. boosting vision) to mitigate language prior hallucinations. In contrast, our diagnostic analysis reveals that in RAG, *useful retrieved text itself* leads to attention distraction. To address this, MAD-RAG introduces a mechanism of a fundamentally different granularity: **intra-text decoupling** to form a dual-question strategy (Eq. 2) and selectively preserves image-conditioned evidence through attention mixing (Eq. 3–5) based on causality (Sec. 5.1, lines 197-208) and recency bias in decoding-based LLMs  (Sec. 5.2, lines 256-264). **To the best of our knowledge, we believe we are the first to identify such a previously unexplored failure mode in retrieval-augmented LVLMs and propose a framework to solve it.** Reviewers guYh, yHK6, and rqPV explicitly recognize this as "a valuable perspective," "highly important," and "novel," respectively. Additionally, as shown in **Table 1**, simply adapting prior methods like VCD to our setting shows **limited or negative gains**, proving that a structurally distinct intervention is required.
> > >
> > > We hope this clarifies that our contribution stems from **new findings and diagnosis** and introduces a **structurally distinct intervention** suited to them.

---

### Official Review · Reviewer_yHK6 · 2026-03-13

**Soundness:** 3
**Presentation:** 3
**Significance:** 4
**Originality:** 4
**Overall Recommendation:** 5
**Confidence:** 5

**Summary:**

This paper investigates the failure mode of retrieval-augmented generation for VLMs, where the model still produces incorrect answers even when the retrieved context is accurate. The authors attribute this phenomenon to attention distraction, which manifests in two forms: cross-modal and intra-image attention distraction.

To address this issue, this paper proposes MAD-RAG, a training-free method that enhances vanilla RAG through two lightweight modifications.
 (1) Dual-Question Format: MAD-RAG duplicates the input question to construct two identical question token groups, $Q_I$ and $Q_C$, which attend to visual tokens and context tokens, respectively.
 (2) Attention Mixing: MAD-RAG further transfers image-grounded attention from $Q_I$ to $Q_C$ at each layer of the VLM.

Extensive experiments demonstrate the effectiveness of MAD-RAG, which outperforms both existing RAG-based methods and VLM hallucination mitigation approaches.

**Compliance With Llm Reviewing Policy:**

Affirmed.

**Final Justification:**

The detailed responses have addressed all my concerns. So, I decided to give an Accept with high confidence.

**Key Questions For Authors:**

The key questions the authors should address in the rebuttal are Weaknesses 1, 2,  and 3. Please refer to Strengths And Weaknesses part for more details.

**Limitations:**

yes

**Strengths And Weaknesses:**

### Strengths

- The research question addressed in this paper is highly important. In knowledge-based VQA tasks, it is a common issue that the performance of RAG is sometimes unsatisfactory, even when the ground-truth evidence is provided. This paper investigates this failure mode and proposes a potential explanation: attention distraction.
- The paper presents a detailed and convincing analysis of how retrieval reshapes attention behavior during multimodal generation, identifying both cross-modal and intra-image attention distraction. This analysis provides strong motivation for the proposed methodological design.
- The proposed approach is lightweight and training-free. The efficiency experiments demonstrate that MAD-RAG introduces the lowest computational overhead among all non-trivial baselines, highlighting its practical applicability.
- The paper is well written and well structured, with clear motivation, a detailed task formulation, and comprehensive experimental evaluations.

### Weaknesses

1. MAD-RAG appears to achieve better performance on Qwen2.5-VL-3B than on Qwen2.5-VL-7B. It would be beneficial to include experiments across a wider range of model sizes (e.g., 3B, 7B, 32B, and 72B) to better understand how MAD-RAG's effectiveness scales with model capacity. Furthermore, the current evaluation only includes LLaVA and Qwen2.5-VL. The effectiveness of MAD-RAG should also be validated on additional architectures from different model families.
2. In Figure 5, MAD-RAG fails on some examples that Vanilla RAG answers correctly. Have the authors analyzed the reasons behind this phenomenon?
3. It would be helpful if the authors could visualize the attention heatmaps produced by MAD-RAG, like Figure 1. In particular, a comparison between the attention patterns of MAD-RAG and Vanilla RAG would provide further insight into how the proposed method mitigates attention distraction.

---

> ### Author Rebuttal · Authors · 2026-03-29
>
> We sincerely thank the reviewer for the positive and constructive feedback, and we will incorporate all corresponding revisions into the manuscript.
>
> **Supp link:** https://anonymous.4open.science/r/MAD_RAG/README.md
>
> ----- W1-----
>
> **Why MAD-RAG shows larger gains on Qwen2.5-VL-3B than on 7B.** We thank the reviewer for this observation. We do not interpret this as a strict scaling trend. As shown in Table 1, Qwen2.5-VL-7B already achieves stronger Vanilla RAG performance than Qwen2.5-VL-3B on most benchmarks, which suggests that it may already make better use of retrieved contexts under Vanilla RAG, leaving less room for further recovery by MAD-RAG. In contrast, the 3B model has poorer long-context handling ability, leaving more room for improvement and leading to larger gains on some benchmarks. We will clarify this point in the revision.
>
> **Scaling experiments.** Following the reviewer's suggestion, we include results on Qwen2.5-VL-32B and Qwen2.5-VL-72B:
> | Model | Method | OK-VQA | E-VQA | InfoSeek |
> |-|-|-|-|-|
> | 32B |Vanilla RAG | 53.44 | 84.24|46.20|
> | 32B | **MAD-RAG** | **55.20** | **87.33** | **46.84**|
> | 72B | Vanilla RAG | 69.81 | 83.49 | 49.53 |
> | 72B | **MAD-RAG** | **70.70** | **86.13** |**51.30**|
>
> MAD-RAG consistently improves over the vanilla RAG baseline across all three datasets, indicating that AD is not just a small-model artifact but a general issue affecting LVLMs with RAG.
>
> **Additional architectures.** We include additional experiments in Appendix G (Table 6) to cover more models in **Qwen3-VL** in addition to the results on LLaVA and Qwen2.5-VL in the main text. MAD-RAG consistently outperforms Vanilla RAG across all three benchmarks on those models: on Qwen3-VL-4B, it improves over Vanilla RAG by +1.71/+8.35/+1.17 on OKVQA/E-VQA/InfoSeek, and on Qwen3-VL-8B by +2.16/+6.43/+3.27, respectively. These results demonstrate the generalizability of our approach.
>
> ----- W2-----
>
> **Reasons behind (RAG ✓, MAD-RAG ✗) in Figure 5.** We analyzed residual failure cases where vanilla RAG is correct but MAD-RAG fails (RAG ✓, MAD-RAG ✗). We visualized the closed-book attention maps for these instances and found that the visual attention does not fall on question-relevant objects or regions, indicating inherently weak visual perception for these samples (see Table 11 in **Supp link**). Vanilla RAG's correct answers in these cases are spuriously correct: the model generates the right answer not through visual understanding, but by exploiting language bias in the oracle chunks. Consequently, when MAD-RAG mixes in the closed-book visual attention, it introduces noise from irrelevant image regions into this text-driven shortcut, disrupting the reasoning path. This reveals a clear boundary condition: MAD-RAG's effectiveness is predicated on the LVLM possessing sufficient visual perception for the given query. When this assumption does not hold, there is no correct visual grounding to restore, and the mixing operation can be counterproductive.
>
> **Overall impact.** Importantly, as shown in Table 1 and Figure 5, the number of cases MAD-RAG successfully recovers, particularly the Attention Distraction cases (Closed-book ✓, RAG ✗), far exceeds the cases where it introduces errors, leading to substantial overall accuracy gains across all datasets and model families. The left failures represent a small fraction compared to the improvements, and we believe adaptive alpha design could further reduce these cases, which we leave as a promising direction for future work.
>
>
> ----- W3-----
>
> Thank you for suggesting visualizing MAD-RAG's attention patterns, which would make our claims more intuitive and verifiable. Following your suggestion, we show that attention after using MAD-RAG is redirected from the irrelevant regions back to the question-related visual objects in Table 12 in **Supp link**. We will add a figure in the appendix of our revised manuscript.

---

> > ### Author Rebuttal · Reviewer_yHK6 · 2026-04-02
> >
> > Thank you for the responses. Figure 12 presents good cases that address my concerns about W3.
> >
> > However, I still have some concerns about W1 and W2.
> > 1. For W1, it would be to further evaluate the approach on architectures from different model families, such as InternVL, Phi3V, and GLM-4.5V. Now, the validation in the main text and the Appendix is limited to the LLaVA and QwenVL series.
> > 2. For W2, Figure 11 cannot prove that Vanilla RAG exploits language bias in oracle chunks. I think it would be clearer to show the attention score on the retrieved text.

---

> > > ### Author Response · Authors · 2026-04-08
> > >
> > > Thank you for your valuable feedback and acknowledging that we have addressed partial of your concerns. We sincerely appreciate your further suggestions on W1 and W2. Below, we provide the requested evaluations on different model families for W1 and further clarify with detailed evidence for W2.
> > >
> > > 1. **Further evaluation for W1**.
> > >
> > > Thank you for the suggestion to broaden the model coverage. Following this, we have extended the evaluation to three additional LVLM families: InternVL-3.5, Phi-3V, and GLM-4.5V. The current results consistently show that MAD-RAG improves over vanilla RAG across these architectures (α=0.5).
> > >
> > > | Model | Method | OK-VQA | E-VQA | InfoSeek |
> > > |-|-|-|-|-|
> > > | InternVL-3.5 |Vanilla RAG |62.30|80.96|52.80|
> > > | InternVL-3.5 | **MAD-RAG** |**63.92**|**84.88**|**54.33**|
> > > | Phi3V | Vanilla RAG |65.66|76.13|52.06|
> > > | Phi3V | **MAD-RAG** |**66.76**|**79.81**|**52.91**|
> > > | GLM-4.5V | Vanilla RAG     |58.56    |85.68       |53.57|
> > > | GLM-4.5V | **MAD-RAG**  |**59.57**|**86.86**|**53.74**|
> > >
> > > Combined with the LLaVA and Qwen-VL-2.5/3 results in our manuscript, the results show that our methods are robust to diverse model families. We will include the new results in the revised appendix G.
> > >
> > >
> > > 2. **Clarification for W2.**
> > >
> > > Thank you for your valuable insights. Your feedback has helped us realize that our original presentation was not as clear as it should have been, and we sincerely appreciate the chance to address this properly.
> > >
> > > **Fig. 11's intended role.** We apologize for any confusion the original framing may have caused.  We would like to clarify that Fig. 11 is intended to present the failure cases of MAD-RAG. When the close-book visual attention is **not on the question-related regions**, mixing the closed-book attention can introduce noise, leading to incorrect predictions in MAD-RAG.
> > >
> > > **Clarification on language bias and new evidence (following your suggestion).** We referred language bias as *“the tendency of models to prioritize language patterns or prior knowledge over the actual visual context presented in the input”* as defined in [1] and observed that there are cases where the model can directly rely on the retrieved chunks and ignore images (e.g., without image input) to make correct predictions (similar to [1]). In those cases, vanilla RAG may slightly outperform MAD-RAG. To directly address your concern, we analyzed the attention score over the retrieved text with the image removed. As shown in **Fig. 13, Supp link** (https://anonymous.4open.science/r/MAD_RAG/README.md), the model assigns **highest attention score to answer-supporting words/phrases** without seeing the image, indicating that in these cases, the prediction can be driven by language bias in the retrieved chunks. We hope this additional analysis helps resolve the concern, and we are very grateful for the suggestion that prompted it.
> > >
> > > **The successful recoveries substantially outweigh the failures.** Importantly, while a small number of cases MAD-RAG did not outperform vanilla RAG, as shown in Table 1 and Fig. 5, the number of cases MAD-RAG successfully recovers far exceeds the cases where vanilla RAG introduces errors, leading to substantial overall accuracy gains across all datasets and model families.
> > >
> > > We thank you again for the constructive feedback, which has genuinely helped us strengthen both the empirical scope and the clarity of our analysis. We will incorporate the additional results and clarifications in the revised appendix.
> > >
> > > [1] He, Jinghan, et al. "Cracking the code of hallucination in lvlms with vision-aware head divergence." ACL, 2025.

---

### Official Review · Reviewer_rqPV · 2026-03-13

**Soundness:** 3
**Presentation:** 3
**Significance:** 3
**Originality:** 3
**Overall Recommendation:** 4
**Confidence:** 3

**Summary:**

This paper identifies Attention Distraction (AD) as a previously overlooked failure mode in retrieval-augmented Large Vision-Language Models, where retrieved text suppresses visual attention globally and shifts focus away from question-relevant image regions, causing failures on questions answerable without retrieval. The authors propose MAD-RAG, a training-free intervention that decouples visual grounding from context integration via a dual-question formulation and preserves image-conditioned evidence through attention mixing. Extensive experiments on OK-VQA, E-VQA, and InfoSeek demonstrate that MAD-RAG rectifies up to 74.68% of RAG-induced failures with negligible computational overhead, outperforming existing baselines across diverse model families.

**Compliance With Llm Reviewing Policy:**

Affirmed.

**Key Questions For Authors:**

1. Adaptive α selection: Given that different datasets (OK-VQA vs. E-VQA) require different α values, can α be selected adaptively through uncertainty-based dynamic adjustment or lightweight meta-learning, rather than manual tuning and grid search?
2. Extension to closed-source models: For commercial APIs without attention access, what specifically are the "prompt or output-level approximations" mentioned? Are there preliminary experiments validating these methods can reproduce attention mixing effects?
3. Trigger conditions of AD: Is AD specific to certain types of retrieved content (e.g., highly relevant but distracting text) or question difficulty? Can fine-grained analysis of failure cases reveal when AD is most likely to occur, enabling more precise risk detection?

**Limitations:**

No. Despite briefly mentioning future work on adaptive α selection and closed-source extensions, the paper inadequately discusses more potential limitations.

**Strengths And Weaknesses:**

Strengths：
1. The paper identifies Attention Distraction (AD) as a novel failure mode in retrieval-augmented LVLMs, challenging the conventional wisdom that RAG failures stem from insufficient attention to retrieved context, and revealing the counter-intuitive phenomenon where retrieved text suppresses visual attention, providing significant diagnostic value.
2. The proposed MAD-RAG method is elegantly designed and practical, decoupling visual grounding from knowledge integration via a dual-question formulation and restoring question-relevant visual evidence through attention mixing, requiring no training and introducing minimal computational overhead.
3. The experimental design is comprehensive and rigorous, covering diverse architectures including LLaVA-1.5, Qwen2.5-VL, and Qwen3-VL across three knowledge-intensive VQA benchmarks, with thorough analysis of both cross-modal and intra-image attention distraction phenomena.

Weaknesses：
1. The attention mixing weight α lacks an adaptive selection mechanism; optimal values vary across datasets, requiring manual tuning.
2. The method relies on internal attention intervention, making it difficult to apply directly to closed-source commercial models; the paper only mentions future work involving prompt-based approximations without providing concrete solutions.
3. Insufficient theoretical analysis regarding the underlying mechanisms of why retrieved text causes attention distraction, with weak theoretical justification for why the dual-question formulation effectively alleviates AD.

---

> ### Author Rebuttal · Authors · 2026-03-29
>
> ----- Q1/W1 -----
>
> We thank the reviewer for this insightful question. We note that **Reviewer guYh, W1** raised a similar question regarding adaptive α strategies. Due to space constraints in individual responses, we provide our full analysis in our response to **Reviewer guYh, W1**, including experimental results for four adaptive α approaches (context-length-based, output confidence-based, majority voting, and retrieval confidence-based). We kindly refer the reviewer there for the complete details.
>
> In brief, our findings show that simple adaptive strategies do not reliably outperform the fixed α, while majority voting across multiple α values approaches, some methods slightly improve but incurs increased inference cost (e.g. 9x). We acknowledge that potentially there will be a better strategy to find alpha via meta-parameter training, but in this work, we focus on training-free and practical recipes and leave this training as future work.
>
>
> -----Q2/W2 -----
>
> We conduct preliminary experiments using a two-pass prompting strategy to approximate attention mixing for closed-source models at the prompt level. In Pass 1, we provide [I, Q], prompting the model to point out the most relevant visual target. In Pass 2, we insert the identified target as an explicit "look at {target}" directive (denoted as L) into the prompt alongside the context and question ([I, Q_I, C, L, Q_C]) to mimic the effect of mixing attention. We evaluate on OKVQA using Gemini-3.0-Flash:
>
> |Setting|Input|Gemini-3.0-flash|
> |-|-|-:|
> |Closed-book|[I, Q]|26.92|
> |Vanilla RAG|[I, Q, C]|35.10|
> |Dual-question w/o attention mixing|[I, Q_I, C, Q_C]|40.11|
> |**MAD-RAG**| [I, Q_I, C, L, Q_C]|**44.39**|
>
> This provides a potential prompt-based approximation for MAD-RAG on closed-source models.
>
> ----- W3 -----
>
> We thank the reviewer for this important feedback. We would like to clarify that this work primarily focuses on **mechanism and empirical insight**, as stated in our contributions. Our primary contribution is to identify Attention Distraction (AD) as a previously overlooked failure mode in retrieval-augmented LVLMs. Prior work largely attributes failures to retrieval quality or knowledge conflicts, whereas we show that degradation can occur **even with accurate and relevant context**, indicating a distinct underlying mechanism. We also identify two forms of AD via empirical study: (1) Cross-modal attention distraction; and (2) Intra-image attention distraction. We are the first to identify and systematically characterize AD in retrieval-augmented LVLMs, supported by quantitative attention analysis, attention heatmap visualization, cross-model generalization, and recovery analysis (Fig. 2) that causally isolates AD from other failure modes.
>
> While we provide a mechanistic explanation grounded in causal masking in attention dynamics  (Sec. 5.1), amplified recency bias leads to the distraction of visual tokens (Sec. 5.2), and Tables 4 empirically validate each component's necessity, we agree that a more formal theoretical analysis would further strengthen the work. We therefore position this as an important direction for future research.
>
> ----- Q3 -----
>
> We thank the reviewer for this insightful question. We perform the analysis of when AD is most likely to occur.
>
> **Retrieval quality-based study.** Table 3 shows two clear trends. (1) As chunks move from top-1 to top-5, the context becomes longer, MAD-RAG yields larger recovery gains. This indicates AD is more prone to occur when the context is longer. (2) The recovery gains under CLIP-L are consistently larger than those with oracle chunks, suggesting that AD is more likely to occur with poor retrievers, where retrieved content seems to be relevant but distracting. This makes robust methods such as MAD-RAG particularly important in low quality retrieval settings.
>
> **Difficulty-based study.** For each sample, we examine the closed-book predictions on the OKVQA dataset. We define a sample as easy if at least two of the four models answer it correctly, and as hard if at most one model answers it correctly. The AD (Closed-book ✓, vanilla RAG ✗) statistics on two difficulty levels is shown below:
>
> |Model|Easy (n=3571)|Hard (n=1475)|
> |-|-:|-:|
> |LLaVA-1.5-7B| 422 (11.82%)|57 (3.86%)|
> |LLaVA-1.5-13B| 386 (10.81%)|58 (3.93%)|
> |Qwen2.5-VL-3B| 433 (12.13%)|76 (5.15%)|
> |Qwen2.5-VL-7B| 438 (12.27%)|74 (5.02%)|
>
> We observe a clear and consistent trend that AD occurs much more often on easy questions than on hard ones across all four models. This is reasonable, since the condition of AD is that the base model can answer correctly (Closed-book ✓) with parametric knowledge, but provides incorrect answers after using RAG. These results identify easy questions as a high-risk regime where robust methods like MAD-RAG are especially useful.

---

> > ### Author Rebuttal · Reviewer_rqPV · 2026-04-07
> >
> > Thank the authors for their great efforts in addressing all my concerns. Most of my concerns and questions have been resolved.

---

> > > ### Author Response · Authors · 2026-04-08
> > >
> > > We sincerely thank the reviewer for the positive feedback and for acknowledging our efforts in the rebuttal. We are pleased to know that most of the concerns have been addressed. If there are any remaining issues or points requiring clarification, we would be more than happy to provide further explanation.

---

### Decision · Program_Chairs · 2026-04-30

**Decision:**

Accept (regular)

**Comment:**

After all reviewers have acknowledged the rebuttal, this paper received one weak reject, one weak accept, and two accept. Reviewers are satisfied with the strengths, such as the clear motivation, interesting idea, sufficient and good results. The rebuttal addresses most of the concerns. I think the current manuscript is not ready for publication.